# Algorithmic Stability and Generalization of an Unsupervised Feature Selection Algorithm

**Xinxing Wu, Qiang Cheng**[*]
University of Kentucky, Lexington, Kentucky, USA

## Abstract

Feature selection, as a vital dimension reduction technique, reduces data dimension by identifying an essential subset of input features, which can facilitate interpretable insights into learning and inference processes. Algorithmic stability is a key characteristic of an algorithm regarding its sensitivity to perturbations of input samples. In this paper, we propose an innovative unsupervised feature selection algorithm attaining this stability with provable guarantees. The architecture of our algorithm consists of a feature scorer and a feature selector. The scorer trains a neural network (NN) to globally score all the features, and the selector adopts a dependent sub-NN to locally evaluate the representation abilities for selecting features. Further, we present algorithmic stability analysis and show that our algorithm has a performance guarantee via a generalization error bound. Extensive experimental results on real-world datasets demonstrate superior generalization performance of our proposed algorithm to strong baseline methods. Also, the properties revealed by our theoretical analysis and the stability of our algorithm-selected features are empirically confirmed.

## 1 Introduction

High-dimensional data is challenging due to the curse of dimensionality [7]. Dimensionality reduction is an important technique for dealing with such data, comprising two typical approaches: feature extraction and feature selection. The former, including principal component analysis (PCA) [35] and autoencoder (AE) [41, 5], is widely used in various fields such as biology [3] and computer vision [49, 18, 38]. Nonetheless, new features produced by feature extraction form a new space and, in general, do not have a direct correspondence to original features, leading to difficulty in deriving interpretable insights for the domain problems, such as biomarker identification and drug discovery. Alternatively, feature selection identifies essential features from the original feature space, providing critical interpretations and insights in many tasks [10, 45], e.g., gene functional enrichment analysis, biomarker detection, and high-throughput screening for drug discovery [22, 47, 27].

Stability is an important characteristic of an algorithm, which quantifies the sensitivity of the output to the perturbation of its training samples [12, 19]. The current stability analyses of feature selection algorithms mainly focus on similarity-based and frequency-based stability measures for practical assessment, e.g., [16, 21, 34], rather than the algorithms themselves, leaving feature selection with proper algorithmic stability [8] an unmet need. Especially, existing unsupervised feature selection algorithms use the empirical error, or its proxy, to represent generalization error but provide no theoretical guarantee of such use, leaving their ability to generalize to new data unclear.

To address these issues, in this paper we propose a novel unsupervised feature selection algorithm with a proven algorithmic stability guarantee. Our approach trains a neural network (NN) to globally score all features and a dependent sub-NN to select the features with the highest scores to reconstruct the

---

[*]Correspondence should be addressed to: qiang.cheng@uky.edu.

35th Conference on Neural Information Processing Systems (NeurIPS 2021).

original data. Mathematically, we analyze our new algorithm and provide a generalization error upper bound, ensuring its algorithmic stability for guaranteed learning performance. Our contributions are summarized in the following:

- We propose an innovative approach for feature selection. It constructs a NN with a feature scorer to globally score all the features and a dependent sub-NN with a feature selector to locally evaluate the representation ability of highly scored features. Thus, our algorithm is capable of both globally exploring and locally excavating essential features.

- We establish performance guarantees for our algorithm, including a proven convergence rate $\mathcal{O}\left(1/\left(n\min\{\sqrt{\lambda_1}, \lambda_1\}\right) + 1/n\right)$ to ensure uniform stability and a rate $\mathcal{O}\left(1/\left(\sqrt{n}\min\{\sqrt{\lambda_1}, \lambda_1\}\right) + 1/\sqrt{n}\right)$ for generalization error. Here, $n$ is the number of training samples, and $\lambda_1$ is a regularization parameter.

- We confirm the effectiveness of our proposed algorithm with extensive experiments on 10 real datasets. It achieves more competitive performance for data reconstruction and downstream classification tasks than state-of-the-art methods. Notably, the features selected by our algorithm have performance comparable to the original features. Further, the properties revealed by our theoretical analysis and the stability of our algorithm-selected features are empirically verified.

The remainder of the paper is organized as follows. We first discuss the related work, then present our proposed algorithm. Next, we show that our proposed algorithm is uniformly stable and has a guaranteed generalization bound. Finally, we conduct extensive experiments to validate our proposed new algorithm and the properties related to our proved generalization bound.

## 2    Related work

In the literature, the approaches for bounding generalization error can be generally grouped into two categories: One by controlling the complexity of hypothesis spaces and the other by focusing on the property of learning algorithms. The former usually need to define an additional complexity measure on the hypothesis space, such as the Vapnik-Chervonenkis dimension [44] and the Rademacher complexity [6]; the latter mainly include algorithmic stability [8] and robustness-based analysis [50]. In this paper, we focus on algorithmic stability to analyze the stability and generalization of our proposed new algorithm.

**Algorithmic stability in learning theory.**    Algorithmic stability is an important tool for analyzing the generalization of algorithms in learning theory. It characterizes the sensitivity of the loss when the inputs to an algorithm are changed. Simply speaking, if an algorithm is stable, then its loss does not change significantly when the training samples are modified slightly, such as deleting and replacing a sample. Since the inception of the notion [42, 43], the concepts and properties of stability have become a significant learning theory topic. Bousquet and Elisseeff [8] have shown that stability has a direct connection with generalization in that the uniform stability of a learning algorithm implies a tight generalization error bound. Thanks to this connection, algorithmic stability has been widely used for generalization analysis of learning algorithms, including regularized least squares regression [8], multi-class support-vector-machine classification [39], multi-task learning [28], supervised autoencoders [23], and distributed learning [46]. Especially in [28], the auxiliary tasks are used for regularization, and so is the reconstruction [23]. It is noted that all these studies are on supervised learning. For unsupervised learning, stability analysis of algorithms is yet to be developed; particularly, the role of regularization in the stability bound remains elusive.

**Stability studies of feature selection algorithms.**    A few existing studies about the stability of feature selection algorithms are mainly on stability measures for quantifying the sensitivity to input changes. In [21] a Shannon entropy-based stability measure is proposed for evaluating the stability of selected features. In [16] the stability of a feature selection algorithm is assessed by comparing the average pairwise similarity of all feature subsets obtained from different subsamples by the algorithm. The recent work [34] generalizes the requirements of stability measures into five properties and proposes a statistical estimator for stability to satisfy these properties. Although able to provide relevant assessment information about the selected features to some extent, most stability measures have no essential connection with the corresponding feature selection algorithms. In addition, they

are demonstrated empirically in general, without theoretical insights and generalization guarantees. The only exception appears to be the theoretical proof for uniform weighting stability [25]. This work builds an algorithm called feature weighting as regularized energy-based learning (FREL), proves stability, and extends to a further ensemble version. FREL is for supervised feature selection, and the uniform weighting stability does not appear to be linked with generalization. In brief, despite the wide use of algorithmic stability in analyzing many learning algorithms, its use and analysis for unsupervised feature selection algorithms are still unmet needs.

**Unsupervised feature selection algorithms.** Many unsupervised feature selection algorithms have been proposed in recent years to meet the demand for effectively handling large-scale and high-dimensional data. They can be generally classified into four categories: filter, wrapper, embedder, and hybrid approaches [2, 24]. In this paper, we will mainly compare our algorithm with ten typical unsupervised feature selection algorithms, including Laplacian score (LS) [17], principal feature analysis (PFA) [30], SPEC [52], multi-cluster feature selection (MCFS) [9], unsupervised discriminative feature selection (UDFS) [51], nonnegative discriminative feature selection (NDFS) [26], Autoencoder feature selector (AEFS) [15], agnostic feature selection with slack variables (AgnoS-S) [11],[2] graph-based infinite feature selection (Inf-FS) [40], and concrete autoencoders (CAE) [1]. LS, SPEC, and Inf-FS are filter approaches; PFA can be classified as a wrapper approach; the other six are embedded algorithms. Among them, AEFS, AgnoS-S, and CAE are AE-based feature selection methods: AEFS combines AE regression and $\ell_{2,1}$ regularization on the weights of the encoder to obtain a subset of useful features; AgnoS-S adopts AE with the $\ell_1$ norm on slack variables in the first layer of AE to implement feature selection; CAE replaces the first hidden layer of AE with a concrete selector layer [31], and then it selects the features with a high probability of connection to the nodes of the concrete selection layer. While frequently used, these methods have no theoretical analysis of their algorithmic stability and leave their generalization abilities to new data unclear. Indeed, they generally lack such needed abilities, as demonstrated in our experiments (see Table 3).

## 3 Notations and preliminary

Let $n$, $m$, $k$, and $d$ be the numbers of training samples, features, selected features, and reduced dimensions. Let $X \in \mathbb{R}^{n \times m}$ be a sample matrix, and $X^T$ be its transposition. A lowercase capital letter like $w_m$ denotes a vector; $\text{Diag}(w_m)$ represents a diagonal matrix with the diagonal $w_m$. $w_m^{\max_k}$ stands for the operation to keep the $k$ largest entries of $w_m$ while making the other entries 0. Let $\phi : \mathbb{R}^m \to \mathbb{R}^m$ be an element-wise operation. $\| \cdot \|_F$, $\| \cdot \|_2$, $\| \cdot \|_1$, and $\| \cdot \|_\infty$ respectively denote the Frobenius, $\ell_2$, $\ell_1$, and infinity norms. The sample space of data is denoted as $\mathcal{X}$. We assume all samples are bounded, i.e., $\forall x \in \mathcal{X}, \exists \kappa_1 > 0$, such that $\|x\|_2 \leqslant \kappa_1$. Besides, $\forall x, x' \in [0, \infty)$, for a function $f$, $\Delta^t(f(x), x') \triangleq f(x) - f(x + t\Delta x)$, where $\Delta x = x' - x$ and $t \in [0, 1]$.

Let $S \triangleq \{x_i \in \mathcal{X}, i = 1, 2, \ldots, n\}$ be a finite set of training samples which are independently and identically distributed according to an unknown distribution $P$. We denote the sets after removing and replacing the $i$-th element from $S$ by $S^{\backslash i} \triangleq \{x_1, \ldots, x_{i-1}, x_{i+1}, \ldots, x_n\}$ and $S^i \triangleq \{x_1, \ldots, x_{i-1}, x'_i, x_{i+1}, \ldots, x_n\}$, respectively. Let $\boldsymbol{A}$ be an algorithm, and $\boldsymbol{A}_S$ be a function picked by $\boldsymbol{A}$ from the hypothesis space $\mathcal{H}$ based on $S$. $\boldsymbol{A}$ is assumed to be deterministic and symmetric with respect to $S$. Let $\ell : \mathcal{X} \times \mathcal{X} \to [0, +\infty)$ be a loss function. The generalization error is

$$L(\boldsymbol{A}, S) \triangleq \mathbb{E}_x \left[ \ell(\boldsymbol{A}_S, x) \right] = \int_{\mathcal{X}} \ell(\boldsymbol{A}_S, x) dP,$$

and the empirical error is

$$L_{emp}(\boldsymbol{A}, S) \triangleq \frac{1}{n} \sum_{i=1}^n \ell(\boldsymbol{A}_S, x_i),$$

where $x_i \in \mathcal{S}, i = 1, 2, \ldots, n$.

To study the stability of $\boldsymbol{A}$, the leave-one-out error is used,

$$L_{loo}(\boldsymbol{A}, S) \triangleq \frac{1}{n} \sum_{i=1}^n \ell(\boldsymbol{A}_{S^{\backslash i}}, x_i).$$

---

[2]In [11], three variants of AE-based agnostic feature selection algorithms are proposed. AgnoS-S is generally the best among the three; thus, in this paper we will compare our algorithm with AgnoS-S.

By the empirical and leave-one-out errors, uniform stability is defined as follows:

**Definition 1** (Uniform Stability [8]). *An algorithm $\boldsymbol{A}$ has uniform stability $\beta$ with respect to $\ell$ if $\forall S \in \mathcal{X}^n, i = 1, 2, \ldots, n$,*

$$\|\ell(\boldsymbol{A}_S, \cdot) - \ell(\boldsymbol{A}_{S \setminus i}, \cdot)\|_\infty \leqslant \beta,$$

*where $\beta$ is a function of $n$. Generally, for a uniformly stable algorithm, $\beta$ decreases as $\mathcal{O}(1/n)$.*

## 4 Method

Now we will present our novel AE-based feature selection algorithm.

**Revisit of AE.** For AE, we formalize it as follows:

$$\min_{f,g} \|\mathrm{X} - f(g(\mathrm{X}))\|_\mathrm{F}^2,$$

where $g$ is an encoder, and $f$ is a decoder. $g(\mathrm{X})$ embeds the input data into a latent space $\mathbb{R}^{n \times d}$, where $d$ denotes the dimension of the bottleneck layer.

**Formalization of unsupervised feature selection.** The goal of feature selection is to identify a subset of important features in the original feature space, and it can be formalized as follows:

$$\min_{S(k),H} \|H(\mathrm{X}_{S(k)}) - \mathrm{X}\|_\mathrm{F}^2, \tag{1}$$

where $S(k)$ denotes a subset of $k$ features of X with $k < m$, $\mathrm{X}_{S(k)}$ is the resulting dataset by restricting X to the selected subset. We use $H$ to represent a mapping on the $k$-dimensional space, and it selects a subset of the features of X to preserve the information of X as much as possible. The optimization problem of evaluating the subset of features is typically NP-hard [32, 14]. This paper will develop an efficient algorithm to effectively approximate the solution of (1).

**AE-based new algorithm for feature selection.** Our model is constructed as follows:

$$\min_{\mathrm{W_I},f,g} \quad \|\mathrm{X} - f(g(\mathrm{X}(\Phi(\mathrm{W_I})^{\max_k})))\|_\mathrm{F}^2 + \lambda_1 \|\mathrm{X} - f(g(\mathrm{X}(\Phi(\mathrm{W_I}))))\|_\mathrm{F}^2, \tag{2}$$

where $\Phi(\mathrm{W_I}) \triangleq \mathrm{Diag}(\phi(\mathrm{w}_m)) \in \mathbb{R}^{m \times m}$, $\Phi(\mathrm{W_I})^{\max_k} \triangleq \mathrm{Diag}((\phi(\mathrm{w}_m))^{\max_k}) \in \mathbb{R}^{m \times m}$, and $\lambda_1$ is a regularization parameter. We use NN for the optimization of (2), and the network architecture is shown in Supplementary Figure 1. For the convenience of later discussions, $\forall \mathrm{x} \in \mathcal{X}$, let $\ell^{\mathrm{selec}}(\Phi(\mathrm{W_I})^{\max_k}, \mathrm{x}) \triangleq \|\mathrm{x} - f(g(\mathrm{x}(\Phi(\mathrm{W_I})^{\max_k})))\|_2^2$, and $\ell^{\mathrm{score}}(\Phi(\mathrm{W_I}), \mathrm{x}) \triangleq \|\mathrm{x} - f(g(\mathrm{x}(\Phi(\mathrm{W_I}))))\|_2^2$.

The operators $\Phi(\mathrm{W_I})$ and $\Phi(\mathrm{W_I})^{\max_k}$ are called feature scorer and feature selector, respectively.[3] $\Phi(\mathrm{W_I})$ and $\Phi(\mathrm{W_I})^{\max_k}$ will iteratively interact through the NN and sub-NN. Specifically, (2) can be regarded as an extension of a common NN, with the main difference being the sub-NN, i.e., the first term of (2). During training, $\Phi(\mathrm{W_I})^{\max_k}$ will select the top-$k$ features (from the scorer weights, i.e., $\Phi(\mathrm{W_I})$) by ensuring these selected features to well reconstruct the original input X with the NN. On the other hand, the NN, as a constraint, will ensure the selected features for $\Phi(\mathrm{W_I})^{\max_k}$ to have the largest importances globally. After training, we obtain the trained $\Phi(\mathrm{W_I})^{\max_k}$, and then we use it to make feature selection on new samples during testing. Taking MNIST-Fashion, COIL-20, and USPS as examples, for $k = 50$ and $\Phi^{\max_k} = (\mathrm{W_I^2})^{\max_k}$, we visualize the feature selection and reconstruction results on MNIST demonstrated in Figure 1, and we give the selected features on original samples of USPS in Figure 2.

## 5 Algorithmic stability and generalization

As previously reviewed, existing feature selection methods have not considered algorithmic stability. Generally, they lack this property or generalization ability, as demonstrated in our experiments. Moreover, there has been no clue about the role of regulation in uniform stability. Our algorithm is built to address these issues and, here, it will be shown to possess these desirable properties. For (2),

---

[3]For simplicity, we will use the shorthand notations $\Phi$ and $\Phi^{\max_k}$ to denote $\Phi(\mathrm{W_I})$ and $\Phi(\mathrm{W_I})^{\max_k}$.

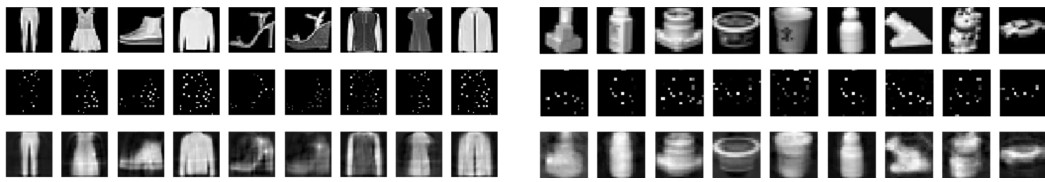

Figure 1: Original testing samples (row 1), 50 selected features (row 2), and reconstruction based on the 50 selected features (row 3) for MNIST-Fashion (left panel) and COIL-20 (right panel). More results are illustrated in Supplementary Material.

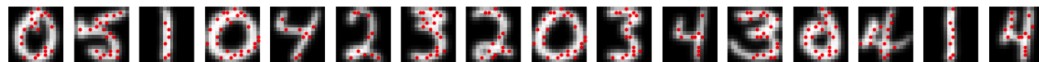

Figure 2: Key features with original samples for USPS.

the first term implies that the original samples are well represented by the selected $k$ features. The second term scores all the features by their reconstruction ability and ensures that the selected features have high scores among all features. It will be used as a regularizer to promote the stability of the feature selection. It requires that all features be explored globally for high representation ability before being further excavated locally by the selector. The role of the scorer will be empirically demonstrated in the experiments (see Figure 3 (b)).

It is worth noting that, in contrast to existing models, such as NNs, which require nonlinear activation functions to capture the nonlinearity, the first term in (2) naturally facilitates a nonlinear model even when the scorer is a linear model. Such a nonlinearity intrinsically comes from ranking the feature weights and selecting the $k$ informative features during the iterative optimization. For simplicity, in the following discussion, we only consider the linear version of $f$ and $g$ in (2), that is, $g(\mathrm{X}) = \mathrm{X}\mathrm{W}_{\mathrm{E}}$, $\mathrm{W}_{\mathrm{E}} \in \mathbb{R}^{m \times d}$, and $f(g(\mathrm{X})) = (g(\mathrm{X}))\mathrm{W}_{\mathrm{D}}$, $\mathrm{W}_{\mathrm{D}} \in \mathbb{R}^{d \times m}$. The experimental section will show that such a concise linear version already achieves superior performance.

**Formalization of stability and generalization.** Let $(\Phi_*^{\max_k}, \Phi_*)$ and $((\Phi_*^{\backslash j})^{\max_k}, \Phi_*^{\backslash j})$ correspond to the optimal feature selectors and feature scorers for the errors

$$RL_{emp}(\Phi, S) \triangleq \frac{1}{n} \sum_{i=1}^{n} \ell^{\mathrm{selec}}(\Phi^{\max_k}, \mathrm{x}_i) + \lambda_1 \ell^{\mathrm{score}}(\Phi, \mathrm{x}_i)$$

and

$$RL_{emp}^{\backslash j}(\Phi, S) \triangleq \frac{1}{n} \sum_{i=1, i \neq j}^{n} \ell^{\mathrm{selec}}(\Phi^{\max_k}, \mathrm{x}_i) + \lambda_1 \ell^{\mathrm{score}}(\Phi, \mathrm{x}_i),$$

where $j \in \{1, \ldots, n\}$. Our goal is to bound $\ell^{\mathrm{selec}}(\Phi_*^{\max_k}, \cdot) - \ell^{\mathrm{selec}}((\Phi_*^{\backslash j})^{\max_k}, \cdot)$, which will be used for the analysis of uniform stability. Further, we intend to bound $L^{\mathrm{selec}}(\Phi_*^{\max_k}, S) - L_{emp}^{\mathrm{selec}}(\Phi_*^{\max_k}, S)$, or $L^{\mathrm{selec}}(\Phi_*^{\max_k}, S) - L_{loo}^{\mathrm{selec}}(\Phi_*^{\max_k}, S)$, to be used in the analysis of generalization errors.

**Assumptions.** To determine the stability and generalization bounds, the following assumptions will be needed:

**Assumption 1.** $\exists \kappa_2 > 0$, such that, $\forall \mathrm{x} \in \mathcal{X}$, $\|(\mathrm{x}(\Phi_*^{\max_k} - (\Phi_*^{\backslash j})^{\max_k})\mathrm{W}_{\mathrm{E}})\mathrm{W}_{\mathrm{D}}\|_2 \leqslant \kappa_2 \|(\mathrm{x}(\Phi_* - \Phi_*^{\backslash j})\mathrm{W}_{\mathrm{E}})\mathrm{W}_{\mathrm{D}}\|_2$.

This assumption is reasonable, because the null space of $\Phi_* - \Phi_*^{\backslash j}$ is smaller than $\Phi_*^{\max_k} - (\Phi_*^{\backslash j})^{\max_k}$; that is, we generally have

$$r\left(\left(\left(\left(\Phi_*^{\max_k} - (\Phi_*^{\backslash j})^{\max_k}\right)\mathrm{W}_{\mathrm{E}}\right)\mathrm{W}_{\mathrm{D}}\right)^{\mathrm{T}}\right) \leqslant r\left(\left(\left(\Phi_* - \Phi_*^{\backslash j}\right)\mathrm{W}_{\mathrm{E}}\mathrm{W}_{\mathrm{D}}\right)^{\mathrm{T}}\right),$$

where $r(\cdot)$ denotes the rank of a matrix. So, it is more likely that the column vector $((\Phi_* - \Phi_*^{\backslash j})\mathrm{W}_{\mathrm{E}}\mathrm{W}_{\mathrm{D}})^{\mathrm{T}}\mathrm{x}^{\mathrm{T}}$ will have $m$ non-zero elements. Consequently, it is more likely that the row vector $(\mathrm{x}(\Phi_* - \Phi_*^{\backslash j})\mathrm{W}_{\mathrm{E}})\mathrm{W}_{\mathrm{D}}$ will have $m$ non-zero elements.

**Assumption 2.** $\exists Z = \{z_1, z_2, \ldots, z_u\} \subset S$, *such that,* $\forall x \in \mathcal{X}$, x *can be reconstructed by* Z, *i.e.,* $x = \sum_{i=1}^{u} \alpha_i z_i + \eta$, *where* $\eta$ *is a small reconstruction error that satisfies* $\|\eta\|_2 \leqslant \kappa_3/n$, *and* $\alpha_i$ *is a scalar that satisfies* $\sqrt{\sum_{i=1}^{u} \alpha_i^2} \leqslant \kappa_4$, *with* $\kappa_3$ *and* $\kappa_4$ *being positive constants.*

This assumption is mild and implies that any sample x can be reconstructed by Z with a small construction error that decreases with $n$.[4] Theoretically speaking, when the samples all reside on a manifold, it obviously holds. Practically, it is related to the self expressiveness of samples exploited in many subspace learning algorithms, e.g., [29, 36, 37].

**Assumption 3.** *Let* $L_{emp}^{score,\backslash j}(\Phi, S) \triangleq (1/n)\sum_{i=1, i\neq j}^{n} \ell^{score}(\Phi, x_i)$ *and* $L_{emp,z}^{score}(\Phi) \triangleq (1/u)\sum_{i=1}^{u} \ell^{score}(\Phi, z_i)$. $\forall t \in [0,1]$,

$$\Delta^t(L_{emp,z}^{score}(\Phi_*), \Phi_*^{\backslash j}) + \Delta^t(L_{emp,z}^{score}(\Phi_*^{\backslash j}), \Phi_*)$$
$$\leqslant \frac{\Delta^t(L_{emp}^{score,\backslash j}(\Phi_*, S), \Phi_*^{\backslash j}) + \Delta^t(L_{emp}^{score,\backslash j}(\Phi_*^{\backslash j}, S), \Phi_*)}{(n-1)/n}. \tag{3}$$

The implication of this assumption is clear: it requires that, at $\Phi_*$ and $\Phi_*^{\backslash j}$, the perturbations of $L_{emp,z}^{score}$ can be controlled by those of $L_{emp}^{score}$, and the left-hand side of (3) is related to Z in Assumption 2. [23] makes a similar assumption; however, we enhance the factor $t$ into $n/(n-1)$ on the right-hand side of (3), which makes Assumption 3 much weaker and more reasonable than that in [23], since $L_{emp,z}^{score}(\Phi) = (n/(n-1))L_{emp}^{score,\backslash j}(\Phi, S)$ when $\mathcal{Z} = S^{\backslash j}$. Further, the perturbations of $L_{emp}^{score}(\Phi_*, S)$ and $L_{emp}^{score,\backslash j}(\Phi_*)$ are implicitly related to $\lambda_1$ (see Proposition 1 below).

**Proposition 1.** *Let* $\Delta\Phi^{max_k} \triangleq (\Phi_*^{\backslash j})^{max_k} - \Phi_*^{max_k}$. $\forall t \in [0,1]$, *the following inequality holds:*

$$\Delta^t(L_{emp}^{score}(\Phi_*, S), \Phi_*^{\backslash j}) + \Delta^t(L_{emp}^{score,\backslash j}(\Phi_*, S), \Phi_*^{\backslash j})$$
$$\leqslant \frac{\left(t\kappa_1^2\left(\left\|\left(2\left(\Phi_*^{\backslash j}\right)^{max_k} - t\Delta\Phi^{max_k}\right)W_E W_D\right\|_2 + 2\right)\|((\Delta\Phi^{max_k})W_E)W_D\|_2\right)}{\lambda_1}.$$

Proposition 1 reveals the relationship of the regularization represented by $\lambda_1$ with the perturbation of $L_{emp}^{score}(\Phi_*, S)$ and $L_{emp}^{score,\backslash j}(\Phi_*, S)$.

As discussed above, we will mainly verify Assumption 2. For this purpose, we develop a core-subspace learning procedure. More details about the verification are given in Supplementary Material.

**Uniform stability.** We establish a bound on uniform stability of our algorithm for (2).

**Theorem 1** (Uniform Stability). *Under Assumptions 1, 2, and 3, we have,* $\forall n \geqslant 2$,

$$\left\|\ell^{selec}(\Phi_*^{max_k}, \cdot) - \ell^{selec}\left(\left(\Phi_*^{\backslash j}\right)^{max_k}, \cdot\right)\right\|_\infty = \mathcal{O}\left(\frac{1}{n\min\{\sqrt{\lambda_1}, \lambda_1\}} + \frac{1}{n}\right). \tag{4}$$

This theorem quantifies the insensitivity of our algorithm to the perturbation of its input, and the optimal feature selector $\Phi_*^{max_k}$ would not change significantly with the change of one sample. The upper bound in (4) is mainly about the approximate behavior of $n$, and it is theoretically more meaningful when $\lambda_1$ is much larger than $1/n$. If $\lambda_1$ is very large, i.e., with over-regularization, then essentially the second term of (2) works, which will make the feature selector underfit training samples and incur large test error. Theoretically, it may still be sufficient for generalization, and the bound in (4) will reduce to $\mathcal{O}(1/n)$; in practice, however, the corresponding validation error should be large. Thus, such a large $\lambda_1$ would never be practically chosen. How to analytically determine the optimal value of $\lambda_1$ is out of the scope of this paper and will be an interesting future research topic. In this paper, we empirically choose a positive value of $\lambda_1$ by using cross-validation (see Experiments for details). If $\lambda_1$ is set to 0, then the second term of (2) will vanish; in this case, the stability with only the first term of (2) warrants further study in the future.

---

[4]Liu et al [28] and Le et al [23] have made a similar assumption.

**Generalization error bound.** Now we will bound $L_{emp}^{\text{selec}}\left(\Phi_*^{\max_k}, S\right) - L^{\text{selec}}\left(\Phi_*^{\max_k}, S\right)$.

**Theorem 2** (Generalization Error). $\exists \kappa_5 > 0$ and $\delta \in (0, 1)$, $\forall \mathrm{x} \in \mathcal{X}$ and $S$, as long as $\ell(\boldsymbol{A}_S, \mathrm{x}) \leqslant \kappa_5$, the following inequality holds with probability at least $1 - \delta$,

$$
L^{\text{selec}}\left(\Phi_*^{\max_k}, S\right) - L_{emp}^{\text{selec}}\left(\Phi_*^{\max_k}, S\right) = \mathcal{O}\left(\frac{\sqrt{\ln\left(\frac{1}{\delta}\right)}}{\sqrt{n}\min\{\sqrt{\lambda_1}, \lambda_1\}} + \sqrt{\frac{\ln\left(\frac{1}{\delta}\right)}{n}}\right).
$$

This theorem shows that, besides the uniform stability bound in Theorem 1, there is an upper bound for the generalization error of our proposed algorithm. By this theorem, as long as our feature selection algorithm is stable, the empirical error has a proven guarantee to approximate generalization error. The convergence rate of generalization error is $\mathcal{O}\left(1/\left(\sqrt{n}\min\{\sqrt{\lambda_1}, \lambda_1\}\right) + 1/\sqrt{n}\right)$. It is noted that, when $\lambda_1 > 1$, the convergence rate is $\mathcal{O}\left(1/\sqrt{n}\right)$, which is different from the rate under the $\ell_2$ regularization; when $\lambda_1 \leqslant 1$, the convergence rate is $\mathcal{O}\left(1/\left(\sqrt{n}\lambda_1\right)\right)$, the same as that under the $\ell_2$ regularization [8]. When the reconstruction error from the feature scorer is made small with a large $\lambda_1 > 1$, the scorer will play a more important role than the selector, and the selector might be underfitted (see Figure 3 (b)); and vice versa for $\lambda_1 \leqslant 1$. Additionally, if we fix $\lambda_1$, the bound will reduce to $\mathcal{O}\left(1/\sqrt{n}\right)$, which decays similarly to the bound under the $\ell_2$ regularization [8] when $\lambda_1$ is fixed.

The proofs of Proposition 1, Theorems 1 and 2, the bound of $L^{\text{selec}}\left(\Phi_*^{\max_k}, S\right) - L_{loo}^{\text{selec}}\left(\Phi_*^{\max_k}, S\right)$, and more discussions are provided in Supplementary Material.

## 6 Experiments

In this section, we will perform extensive experiments to validate our new algorithm.

Table 1: Statistics of datasets.

| No. | Dataset | #Samples | #Features | #Classes | No. | Dataset | #Samples | #Features | #Classes |
|-----|---------|----------|-----------|----------|-----|---------|----------|-----------|----------|
| 1 | Mice Protein | 1,080 | 77 | 8 | 6 | USPS | 9,298 | 256 | 10 |
| 2 | COIL-20 [33] | 1,440 | 400 | 20 | 7 | GLIOMA | 50 | 4,434 | 4 |
| 3 | Activity [4] | 5,744 | 561 | 6 | 8 | Prostate_GE | 102 | 5,966 | 2 |
| 4 | ISOLET | 7,797 | 617 | 26 | 9 | SMK_CAN_187 | 187 | 19,993 | 2 |
| 5 | MNIST-Fashion [48] | 10,000 | 784 | 10 | 10 | arcene | 200 | 10,000 | 2 |

**Datasets to be used.** The benchmarking datasets and their statistics are summarized in Table 1.[5] Following CAE [1] and considering the long runtime of UDFS, for dataset 5, we randomly choose $6,000$ samples from the training set for training and validating and $4,000$ samples from the test set for testing. We then randomly split $6,000$ samples into training and validation sets by a ratio of 90:10. For other datasets, we randomly split the samples into training, validation, and test sets by a ratio of 72:8:20, and we tune hyperparameters on the validation set. More details about these datasets are provided in Supplementary Material.

**Design of experiments.** In all experiments, for our model, we use the same linear decoder as that in [1]. For the encoder, for a fair comparison, our algorithm uses the structure from an AE having one hidden layer with the same number of neurons as other AE-based baselines in comparison. We use the linear activation function for the encoder in the same way as CAE and AEFS in [1]. For feature selection, the first layer of the encoder in our NN uses the slack variables in the same way as AgnoS-S in [11]. We set the maximum number of epochs to 200. We initialize the weights of the feature selection layer by sampling uniformly from $\mathrm{U}[0.999999, 0.9999999]$ and the other layers with the Xavier normal initializer.[6] We adopt the Adam optimizer [20] with a learning rate of $0.001$. We set $\lambda_1$ to $1/2^7$.[7] We take $k = 10$ for dataset 1 and $k = 50$ for datasets 2-6 following CAE [1], and

---

[5]Datasets 1 and 4 are downloaded from http://archive.ics.uci.edu/ml/datasets/. Datasets 6-10 are from the scikit-feature feature selection repository [24].

[6]This distribution makes initial values of the weights in the selection layer close to 1 but still different to break the potential ties.

[7]We tune $\lambda_1$ by searching in $\left\{2, 1/2^0, \ldots, 1/2^{10}\right\}$ on the validation set of MNIST-Fashion, then choose the optimal one and use it on other datasets.

$k = 64$ for high-dimensional datasets 7-10. The dimension of the latent space is consistently set to $k$. The main codes related to our proposed algorithm are publicly available,[8] and the implementation details of baseline algorithms are provided in Supplementary Material.

Two metrics are used for evaluating the performance of algorithms: 1) reconstruction error, which is measured in mean squared error (MSE); 2) accuracy, which is measured by passing the selected features to a downstream classifier as a viable means to benchmark the quality of selected features. For a fair comparison, following CAE [1], after selecting the features, we train an ordinary linear regression model to reconstruct the original features, and the resulting linear reconstruction error is used as metric 1);[9] for metric 2) we adopt extremely randomized trees [13] as the classifier.

**Results on 10 datasets.** Our experimental results on reconstruction and classification with the selected features are displayed in Tables 2 and 3,[10] and $|W_I|$ and $W_I^2$ are shorthands for feature selectors $\Phi^{\max_k} = |W_I|^{\max_k}$ and $\Phi^{\max_k} = (W_I^2)^{\max_k}$. We followed the way of CAE [1] to split samples and to report the final results on the hold-out test set. For a fair comparison, we directly adopt the published results for reconstruction and classification by LS, AEFS, UDFS, MCFS, PFA, and CAE on datasets 1-5 from [1]. From Table 2, it is seen that our algorithm gives smaller linear reconstruction errors than baseline methods on majority datasets, indicating a stronger ability to select a subset of representative features. From Table 3, it is evident that our algorithm exhibits almost consistently superior performance in the downstream classification task on diverse datasets.

Table 2: Linear reconstruction error with selected features by different algorithms.

| Dataset No. | LS | SPEC | NDFS | AEFS | UDFS | MCFS | PFA | Inf-FS | AgnoS-S | CAE | Ours $|W_I|$ | Ours $W_I^2$ |
|---|---|---|---|---|---|---|---|---|---|---|---|---|
| 1 | 0.603 | 0.051 | 0.041 | 0.783 | 0.867 | 0.695 | 0.871 | 0.601 | 0.013 | 0.372 | 0.009 | **0.008** |
| 2 | 0.126 | 0.413 | 0.134 | 0.061 | 0.116 | 0.085 | 0.061 | 0.130 | 0.038 | 0.093 | **0.015** | 0.016 |
| 3 | 0.139 | 0.127 | 144.353 | 0.112 | 0.173 | 0.170 | 0.010 | 0.282 | 0.010 | 0.108 | **0.005** | **0.005** |
| 4 | 0.344 | 0.119 | 0.129 | 0.301 | 0.375 | 0.471 | 0.316 | 0.098 | 0.042 | 0.299 | **0.016** | 0.018 |
| 5 | 0.128 | 0.107 | 0.127 | 0.047 | 0.133 | 0.096 | 0.043 | 0.094 | 0.024 | 0.041 | **0.022** | 0.023 |
| 6 | 3.528 | 1.120 | 0.918 | 0.025 | 0.034 | 1.050 | 0.022 | 5.245 | 0.017 | **0.010** | 0.018 | 0.018 |
| 7 | 0.140 | 0.210 | 0.404 | 0.060 | 0.060 | 0.173 | 0.055 | 0.163 | **0.054** | 0.063 | 0.067 | 0.070 |
| 8 | 1.694 | 0.605 | 4.506 | 0.280 | 0.228 | 1.929 | 0.180 | 0.187 | **0.048** | 0.387 | 0.172 | 0.144 |
| 9 | 7.344 | 0.118 | 3.005 | 0.102 | \ | 5.492 | 0.089 | 8.725 | 0.096 | **0.077** | 0.093 | 0.100 |
| 10 | 0.328 | 0.045 | 1335.029 | 0.025 | \ | 4.826 | 0.042 | 329.507 | 0.031 | 0.029 | 0.025 | **0.024** |

Table 3: Classification accuracy (%) with selected features by different algorithms.

| Dataset No. | LS | SPEC | NDFS | AEFS | UDFS | MCFS | PFA | Inf-FS | AgnoS-S | CAE | Ours $|W_I|$ | Ours $W_I^2$ |
|---|---|---|---|---|---|---|---|---|---|---|---|---|
| 1 | 13.4 | 24.5 | 8.3 | 12.5 | 13.9 | 13.9 | 13.0 | 42.6 | 51.9 | 13.4 | 97.2 | **99.1** |
| 2 | 38.9 | 14.9 | 21.2 | 58.0 | 55.6 | 63.5 | 64.2 | 37.8 | 89.2 | 58.6 | 96.9 | **97.9** |
| 3 | 28.0 | 20.3 | 18.8 | 24.0 | 28.7 | 29.5 | 36.4 | 18.1 | 56.1 | 42.0 | **87.7** | 87.6 |
| 4 | 40.7 | 5.8 | 7.3 | 57.6 | 45.5 | 52.2 | 62.2 | 13.7 | 18.1 | 68.5 | **83.9** | 82.4 |
| 5 | 51.7 | 27.6 | 13.8 | 58.0 | 54.7 | 51.3 | 68.3 | 23.5 | 79.1 | 67.7 | **80.8** | 80.5 |
| 6 | 36.3 | 46.3 | 11.9 | 94.2 | 94.4 | 12.6 | 96.0 | 21.0 | 95.6 | 95.5 | **96.1** | 95.4 |
| 7 | 50.0 | 20.0 | 40.0 | 80.0 | 70.0 | 40.0 | 80.0 | 20.0 | 80.0 | 50.0 | **90.0** | 80.0 |
| 8 | 52.4 | 47.6 | 47.6 | 85.7 | 90.5 | 57.1 | 90.5 | 61.9 | 76.2 | 85.7 | **95.2** | 90.5 |
| 9 | 57.9 | 65.8 | 42.1 | 50.0 | \ | 44.7 | 65.8 | 47.4 | 65.8 | **73.7** | 68.4 | 57.9 |
| 10 | 62.5 | 32.5 | 70.0 | 75.0 | \ | 62.5 | 77.5 | 67.5 | 77.5 | 77.5 | **82.5** | 75.0 |
| Average | 43.2±14.1 | 30.5±17.0 | 28.1±19.7 | 59.5±24.7 | 56.7±26.2 | 42.7±17.7 | 65.4±23.5 | 35.4±18.1 | 70.0±21.3 | 63.3±22.4 | **87.9±8.8** | 84.6±11.8 |

Further, we compare the behavior of our algorithm with contemporary algorithms over different $k$. By varying $k$ on ISOLET we obtain the corresponding linear reconstruction errors and classification accuracy rates as the outputs from different algorithms. We plot the linear reconstruction errors in MSE and classification accuracy rates in Supplementary Figure 8. It can be observed that our algorithm demonstrates almost consistently better and more stable performance than feature selection algorithms in comparison.

---

[8]They can be found at https://github.com/xinxingwu-uk/UFS

[9]Here, the reconstruction error denotes the error from the first term of (2).

[10]The "\" mark denotes the case with prohibitive running time, where the algorithm ran for more than a week without getting a result and thus was stopped.

Also, we take $\Phi^{\max_k} = |W_I|^{\max_k}$, and we compare the variation of classification accuracy with the reduction of original features. The results are shown in Supplementary Table 1 and Supplementary Figure 9. It is observed that all the reduction in the number of features is more than $80\%$, while the corresponding reduction in classification accuracy is less than $10\%$ with the exception of ISOLET (which is $11.2\%$). It shows that our algorithm can effectively reduce the number of features but still maintain the classification performance comparable to the original data.

## 7 Discussion

In this section, we will empirically verify the properties related to uniform stability bound and generalization bound in Theorems 1 and 2. Further, we will empirically discuss the algorithmic stability and the stability of selected features and also analyze the time complexity of our algorithm (2).

**Effect of $n$.** We plot the curves of error difference $L^{\text{selec}}(\Phi_*^{\max_k}, S) - L_{emp}^{\text{selec}}(\Phi_*^{\max_k}, S)$ and test error versus $n$ in Figure 3 (a). It is observed that $n$ has a direct effect on the error difference and test error: 1) The more training samples, generally the smaller the test error. This observation verifies that small training sets cause overfitting and, by increasing the training set size, the generalization of our algorithm becomes improved with the test error decreased. 2) The error difference decreases in $n$. This phenomenon is well in line with the theoretical result in Theorem 2.

**Effect of $\lambda_1$.** We plot the curves of the error difference and test error versus $\lambda_1$ in Figure 3 (b). Two observations can be made: 1) The larger $\lambda_1$, the smaller the error difference. Such an experimental result aligns well with Theorem 2. 2) The test error decreases at first and then increases with $\lambda_1$. The reason may be as follows: by increasing $\lambda_1$ from zero to about $0.01$, the feature scorer will play an increasingly important role and facilitate finding appropriate scores for features to reduce the reconstruction error, which helps decrease the test error. The regularization role of the scorer is evident: when $\lambda_1$ increases from $0$ to a small value, the test error decreases. But if $\lambda_1$ increases too much, it would over-emphasize the scorer and the minimization of the reconstruction error, making the feature selector underfit training samples and thus incurring larger test errors. These empirical curves offer informative clues for understanding the role of regularization in the generalization bound and choosing a proper $\lambda_1$ in our experiments.

**Effect of $k$.** We plot the curves of the error difference and test error versus $k$ in Figure 3 (c). It is seen that the larger $k$, the larger the error difference. This observation is in line with Theorem 2 (see Supplementary Material for the detailed expression of upper bound). As discussed below this theorem, the generalization error bound of our algorithm has a similar convergence rate to that under the $\ell_2$ regularization. To see how the selection affects the error difference, we plot the curves of Frobenius norms of $W_E W_D$ and $W_0 W_E W_D$ in Figure 3 (d).[11] Note that they have a similar increase tendency to the error difference. The reason may be that, by selecting more features, $L^{\text{selec}}(\Phi_*^{\max_k}, S)$ and $L_{emp}^{\text{selec}}(\Phi_*^{\max_k}, S)$ would depend on more variables and be more easily affected by the perturbations from the variables; as a result, the feature selector would be more difficult to fit. From Figure 3 (c), we can also observe that the larger $k$, the smaller the test error, which is sensible because more selected features mean less loss of original data and the reconstruction would be better. In addition, from Figure 3 (a)-(c), different $\Phi_*$ scorers, such as $|W_I|$ and $W_I^2$, have similar behaviors, which reflects the tendency implied in Theorem 2.

**Algorithmic stability analysis.** Adopting the same experiment design on MNIST-Fashion in Section 6, we vary $n$ from $3,000$ to $6,000$ with a step size of $1,000$ to obtain different $S$; meanwhile, we delete a sample for each $S$ to get the corresponding $S^{\backslash i}$ and then calculate the left-hand side of (4) on the testing set for these trained models. From the plots in Figure 3 (e)-(f), it is seen that with the increase of $n$, the curve of uniform stability bound presents a downward tendency, which is consistent with our theoretical analysis.

Overall, the above four experiments for interpreting the generalization bound are related to the training sample size $n$, the regularization parameter $\lambda_1$, and the number of selected features $k$, and the stability bound, showing different aspects or behaviors of our algorithm. Taken together, these experiments verify the properties revealed by Theorems 1 and 2.

---

[11]Here, $W_0$ denotes the entries of $W$ are zeros except $k$ ones on its diagonal.

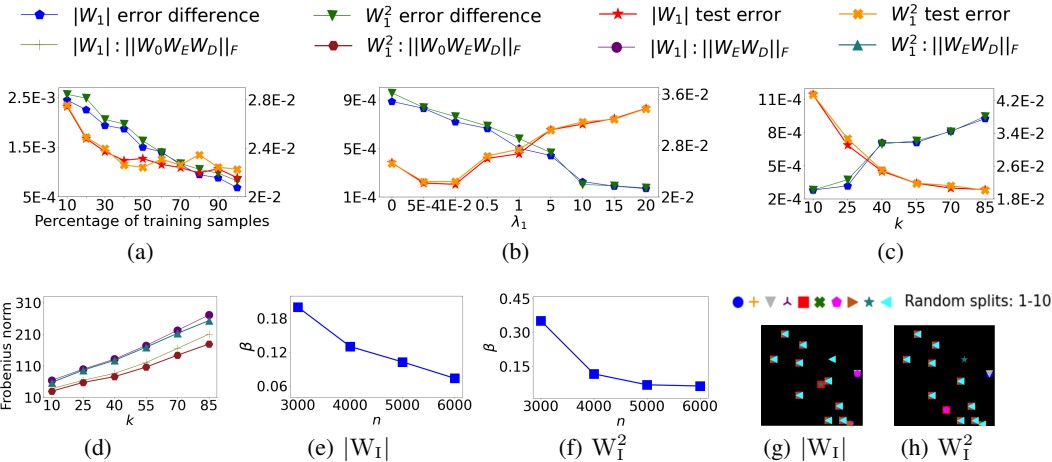

Figure 3: Empirical interpretation of generalization bound and stability analysis on MNIST-Fashion. (a)-(c) show the curves of error difference and test error versus $n$, $\lambda_1$, and $k$, respectively; (d) plots the overall size of network weights versus $k$; (e)-(f) are for algorithmic stability; (g)-(h) are for the stability analysis of 10 selected features. In each (a)-(c), the left vertical axis represents error difference, and the right vertical axis represents test error.

**Stability analysis of selected features.** We empirically analyze the stability of features selected by (2). We randomly split the samples of MNIST-Fashion into the training and testing sets and then use (2) to perform feature selection. We repeat this procedure 10 times with different random seeds and plot the selection results in Figure 3 (g)-(h). Note that the selected features almost overlap for different splits and are stable. More results are provided in Supplementary Material.

**Computational complexity.** Experimentally, the computational time of our algorithm (2) is about twice that of AE. Our algorithm has only an additional sub-NN compared to AE and shares parameters with the NN. Also, the fitting error term of sub-NN is quadratic and similar to that of AE. Therefore, the overall computational complexity of (2) is of the same order as AE.

**Ethical statement.** This paper focuses on feature selection, which is a dimensionality reduction approach. To our best knowledge, there are no ethical issues and negative societal impacts of the proposed technique in this paper.

# 8   Conclusions

In this paper, we propose an innovative unsupervised feature selection algorithm with provable performance guarantees, which consists of a feature scorer and a feature selector. Theoretically, we prove uniform stability and provide the generalization error upper bound for our algorithm. Empirically, we show that our algorithm achieves performance better than the contemporary algorithms on various real-world datasets; additionally, the selected features by the proposed algorithm show comparable performance to the original features. Moreover, we experimentally verify the properties revealed by our theoretical analysis with respect to sample size, regularization levels, the number of selected features, and uniform stability. Additionally, this new algorithm may be applied to other tasks, such as selecting important patterns of image data beyond pixels, which we will extend in future work.

# 9   Funding

This work was partially supported by the NIH grants R21AG070909, R56NS117587, R01HD101508, and ARO W911NF-17-1-0040.

## Acknowledgment

We sincerely thank the anonymous reviewers and AC for their valuable comments.

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
