# Supplementary Material of "Algorithmic Stability and Generalization of an Unsupervised Feature Selection Algorithm"

**Xinxing Wu, Qiang Cheng**[*]
University of Kentucky, Lexington, Kentucky, USA

[*]Correspondence should be addressed to: qiang.cheng@uky.edu.

# 1 Architecture

The architecture of our algorithm is shown in Figure 1. Our feature scorer and selector are not just the direct definitions for the importance of features; instead, the feature scorer and selector will be estimated iteratively through the NN and sub-NN.

Compared with the existing NN architectures, our algorithm has two more hyperparameters, namely $k$ and $\lambda_1$ in Eq. (2) of the main text. In fact, for $\lambda_1$, we can tune it on the validation set. For $k$, it is subject to the practical problem, which is somewhat similar to the number of clusters in $k$-means clustering.

For the training based on Eq. (2) of the main text, in each iteration of backpropagation,

- The selector will require the gradients of the features having the top-$k$ weights in magnitude for this iteration while having no effect on the gradients of other features. Thus, in each iteration, the gradients of the selector need to adopt a ranking operation obtaining the $k$-largest weights from the scorer. And the selector will update the weights of the corresponding $k$ selected features to ensure that these features well reconstruct the original input data X;

- The scorer updates the weights based on the backpropagation, which includes the contribution from the selectors and, at the same time, rescores the features for the selector to perform ranking and selecting the top-$k$ features.

After training, only the trained selector is used to select features and do reconstruction during testing time.

Our design combines global and local considerations for feature selection, which is different from traditional methods for inverse problems, such as Lasso-type methods. In Eq. (2) of the main text, the second term helps obtain $W_I$ to ensure that the selected features are most important and promote the first term to well approximate the input data X. By doing so, theoretically, the algorithmic stability of our proposed algorithm has a guarantee; experimentally, the effect of such a combination is also validated.

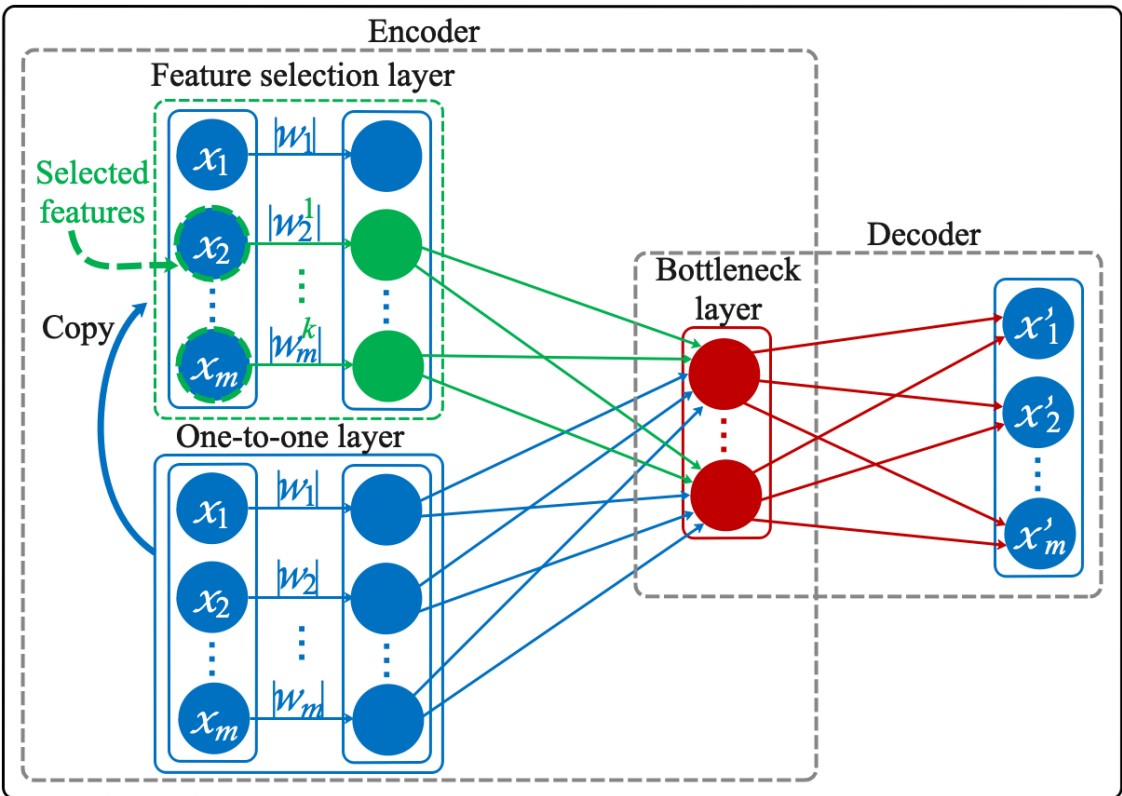

**Figure 1:** The NN's architecture. Illustrated is the feature selector $\Phi(W_I)^{\max_k} = |W_I|^{\max_k}$. During the training phase, the NN (with the one-to-one layer) and its dependent sub-NN (with the feature selection layer) are used to optimize Eq. (2) of the main text. During testing time, only the trained sub-NN is used to select features and do reconstruction.

## 2    Feature Selection and Reconstruction

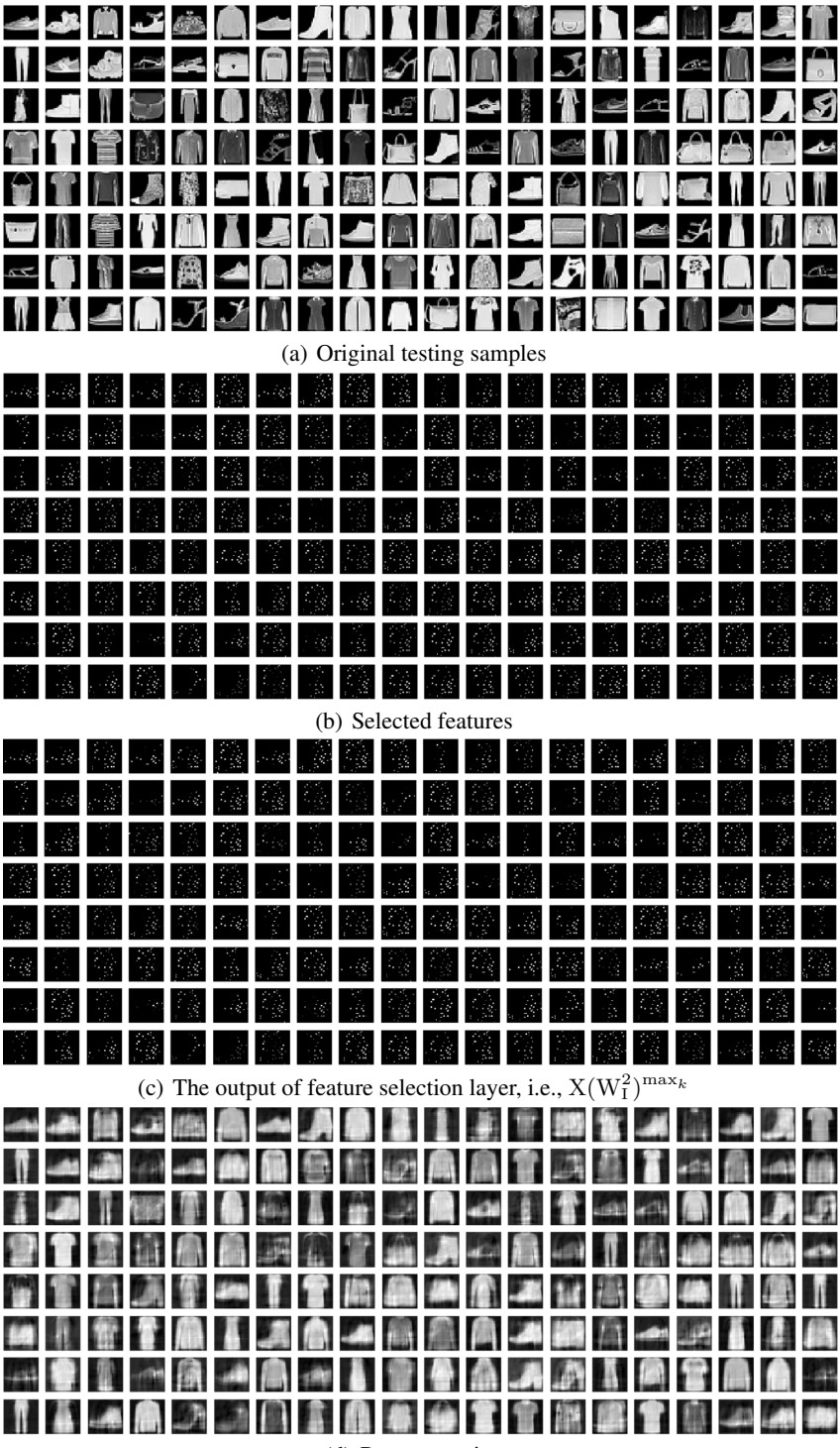

(a) Original testing samples

(b) Selected features

(c) The output of feature selection layer, i.e., $\mathrm{X}(\mathrm{W}_I^2)^{\max_k}$

(d) Reconstruction

**Figure 2:** Original testing samples, 50 selected features, and reconstruction based on 50 selected features for 160 testing samples from MNIST-Fashion. The numbers of epochs and selected features are 200 and 50, respectively. We initialize the one-to-one and feature selection layers' weights by sampling uniformly from $\mathrm{U}[0.999999, 0.9999999]$, and we use the Xavier normal initializer to initialize the other layers' weights.

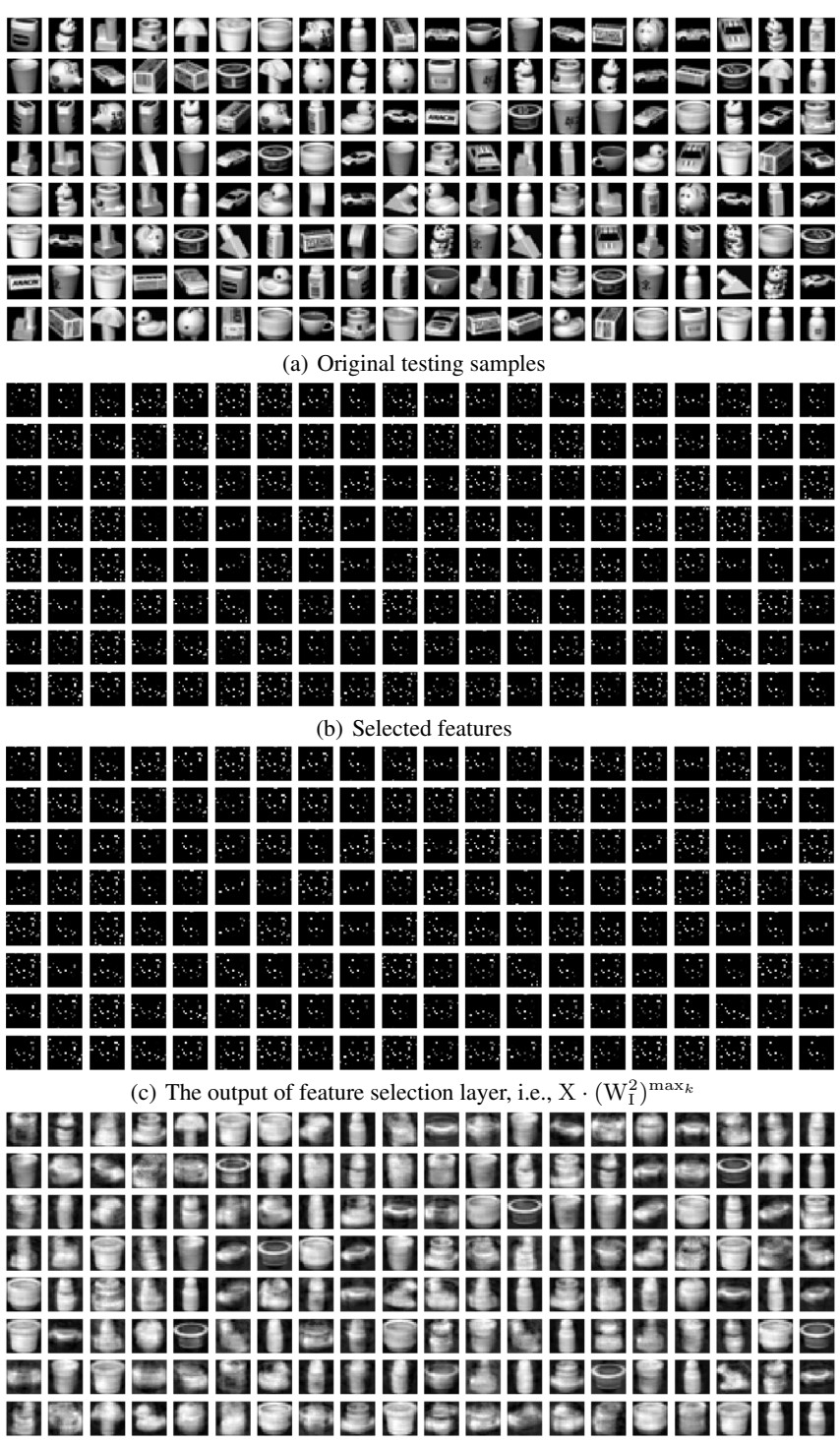

(a) Original testing samples

(b) Selected features

(c) The output of feature selection layer, i.e., $X \cdot (W_1^2)^{\max_k}$

(d) Reconstruction

**Figure 3:** Original testing samples, 50 selected features, and reconstruction based on these selected features for 160 testing samples from COIL-20. The numbers of epochs and selected features are 200 and 50, respectively. We initialize the one-to-one and feature selection layers' weights by sampling uniformly from $U[0.999999, 0.9999999]$, and we use the Xavier normal initializer to initialize the other layers' weights.

## 3  Datasets Used in Experiments

The benchmarking datasets used in this paper are as follows:

**MNIST-Fashion** [Xiao et al., 2017] is a dataset of images, and it has the same image size and structure of training and testing splits as MNIST.

**COIL-20** [Nene et al., 1996] consists of $1,440$ samples. Each sample is a $128 \times 128$ grayscale image. In a similar way to that in CAE [Abid et al., 2019], we resize the original size of images to $20 \times 20$.

**ISOLET** [UCI, 1994] is a dataset of predicting which letter-name was spoken. The features include spectral coefficients, contour features, sonorant features, per sonorant features, and post-sonorant features. The number of samples is 7,797 and the number of features is 617.

**Smartphone Dataset for Human Activity Recognition (HAR) in Ambient Assisted Living (AAL)**[2] [Anguita et al., 2013] were collected from a smartphone worn around the waist of participants when they performed activities such as standing, sitting, lying, walking, walking upstairs, and walking downstairs. It has $5,744$ samples and $561$ features.

**Mice Protein Expression** [UCI, 2015] contains $1,080$ samples. These samples consist of $77$ expression profiles/features, measured in the cerebral cortex of normal and Down syndrome mice.

The following five datasets are taken from the scikit-feature feature selection repository http://featureselection.asu.edu/datasets.php [Li et al., 2017].

**USPS** consists of handwritten digits. It has $7,291$ training and $2,007$ testing images. And the images are $16 \times 16$ grayscale pixels.

**GLIOMA** is a biological dataset. It has $50$ instances, each with $4,434$ features.

**Prostate_GE** is a dataset from medical applications. It has $102$ samples and $5,966$ features.

**SMK_CAN_187** is a dataset consisting of gene expression data from smokers with and without lung cancer. It has $187$ samples, each with $19,993$ features.

**arcene** is a dataset used to distinguish cancer versus normal patterns from mass-spectrometric data. It has 200 samples, each with $10,000$ features.

---

[2]In this study, we refer to it as Activity.

## 4 Design of Experiments Related to Other Algorithms

The implementations of LS, SPEC, NDFS, UDFS, MCFS are taken from the scikit-feature feature selection repository https://github.com/jundongl/scikit-feature.git [Li et al., 2017]. The implementations of AEFS, PFA, and CAE are taken from [Abid et al., 2019]. For UDFS and AEFS, we search the values of regularization hyperparameters and report their results with the optimal hyperparameters in the mean squared error for reconstruction. For other methods, their hyperparameters are set to their default values. Additionally, for CAE, we adopt the linear decoder, i.e., without hidden layers, for datasets 1-6; we use a 1-hidden layer decoder for datasets 7-10 and the comparison of different numbers of selected features on IOSLET. Although we set the maximum number of epochs to $1,000$, to balance the training and validation errors, we use early stopping in optimization.

We implement AgnoS-S according to the settings of [Doquet and Sebag, 2019]: using a single hidden layer, tanh activation for both encoder and decoder, and Adam with an initialized learning rate of $10^{-2}$. For the regularization parameter, it is set to 1. Since the intrinsic dimension is not so easy to obtain practically for some datasets, for a fair comparison, we set the dimension of the bottleneck layer of AE to the number of selected features. The AE weights are also initialized with the Xavier normal initializer as in [Doquet and Sebag, 2019]. For Inf-FS, we adopt the `PyIFS` package[3] in our experiments, where we mainly tune the mixing parameter on validation sets.

In our model, we initialize the weights of the feature selection layer by sampling uniformly from $U[0.999999, 0.9999999]$, the reason is that, by adopting this way of initialization, we can facilitate the selection at the sub-NN and the implementation of the whole model: At the beginning of running our feature selection algorithm, we would like to have all the weights in the selection layer equal because, before incorporating information from the data, all features are regarded as playing equal roles. However, if we initialize all the weights to be 1, then in the first step, the sub-NN could not select the top-$k$ weights; to break the tie, one option is to have the sub-NN randomly select $k$ features in the first step. The adopted initialization is a convenient way to have (almost) equal weights and at the same time avoid random selection in the first step. It may be regarded as a "small trick" for initialization here. Briefly, it is designed to add small perturbations to 1 to help break the potential tie at the beginning.

In our model, $\lambda_1$ is an important regularization parameter. In our paper, we fine-tune $\lambda_1$ on the validation set of MNIST-Fashion, then use the tuned value for other datasets in the spirit of transfer learning. As shown by the experimental results, superior performance can already be achieved. If tuning $\lambda_1$ individually on the validation sets of different datasets, it is expected to achieve better performance; for batch sizes, we individually tune it on the validation sets of different datasets for the case $|W_I|$, then also use them for the case of $W_I^2$.

For the extremely randomized trees, we use the function `ExtraTreesClassifier()` in the library `sklearn.ensemble`. For a fair comparison, we follow the same experimental settings of Abid et al. [2019]: The number of trees in the forest is set to $50$, and other parameters are set to default values. For downstream models, either linear regression or extremely randomized trees, after feature selection, firstly, we split the data with reduced dimensions into training and test data; then, we train these models on the training data and evaluate them on the test data. Such a way of evaluation is commonly used for unsupervised feature selection in the literature, for example, in Abid et al. [2019].

Besides, in the experiments of stability analysis, we use random seeds from 0 to 9; in other experiments, we set the random seed to 0. All the experiments are implemented with Python 3.7.8, Tensorflow 1.14, and Keras 2.2.5. The main codes related to our proposed algorithm are publicly available.[4]

---

[3]https://pypi.org/project/PyIFS/
[4]They can be found at https://github.com/xinxingwu-uk/UFS

## 5 Subspace or Self-Expressiveness-Based Learning

Self-expressiveness of samples is often exploited in many subspace clustering methods, and the similarity matrix C can be obtained by solving the following optimization problem:

$$\min_{C} \left\| X^T - X^T C \right\|_F^2, \tag{1}$$

where $X^T$ is the transposition of X, and $C = \{c_{ij}\}_{n \times n} \in \mathbb{R}^{n \times n}$. $\left\| X^T - X^T C \right\|_F^2$ is the reconstruction error based on self-similarity. The matrix C gives the weights or contributions of different samples in the reconstruction, which can be nominally solved as $(XX^T)^{-1} XX^T$ if $XX^T$ is invertible, but otherwise, no solution exists. Similar to ridge regression, an $\ell_2$-norm regularization term can be added to the objective function of (1) to avoid the ill-defined invertible case, leading to

$$C_{\min} = \arg \min_{C} \left\| X^T - X^T C \right\|_F^2 + \lambda_3 \left\| C \right\|_F^2, \tag{2}$$

where $\lambda_3$ is a positive regularization parameter. The analytic solution of (2) can be obtained as $C_{\min} = (XX^T + \lambda_3 I)^{-1} XX^T$, where I is an identity matrix.

## 6   Detail of Assumption 1

For Assumption 1, we have provided a short explanation below it in the main text. In more detail, if the null space of $\Phi_* - \Phi_*^{\backslash j}$ is smaller than $\Phi_*^{\max_k} - (\Phi_*^{\backslash j})^{\max_k}$, then we have

$$r\left(\left(\left(\Phi_*^{\max_k} - \left(\Phi_*^{\backslash j}\right)^{\max_k}\right) W_E\right) W_D\right) \leqslant r\left(\left(\Phi_* - \Phi_*^{\backslash j}\right) W_E W_D\right).$$

Similarly, we have

$$r\left(\left(\left(\left(\Phi_*^{\max_k} - \left(\Phi_*^{\backslash j}\right)^{\max_k}\right) W_E\right) W_D\right)^T\right) \leqslant r\left(\left(\left(\Phi_* - \Phi_*^{\backslash j}\right) W_E W_D\right)^T\right),$$

where $r(\cdot)$ denotes the rank of a matrix. So, it is more likely that the column vector $((\Phi_* - \Phi_*^{\backslash j})W_E W_D)^T x^T$ will have $m$ non-zero elements. Here, $x^T$ denotes the transposition of $x$. Consequently, it is more likely that the row vector $(x(\Phi_* - \Phi_*^{\backslash j})W_E)W_D$ will have $m$ non-zero elements.

Thus, Assumption 1 is mild and meaningful.

## 7 Verifying Assumption 2

Assumption 2 is mainly about the representative set of feature vectors of samples. Theoretically speaking, when the samples reside on a manifold, Assumption 2 obviously holds. Similar kinds of assumptions for stability analysis are also demonstrated and adopted in [Le et al., 2018] and [Liu et al., 2017].

For practical applications, we may computationally find those $z_i$ based on subspace learning. More specifically, we introduce a procedure, called core-subspace learning, as follows:

$$\min_{C,V_I} \lambda_2 \left\| X^T - \left( X^T V_I^{\max_k} \right) C \right\|_F^2 + \lambda_3 \left\| X^T - \left( X^T V_I \right) C \right\|_F^2 + \lambda_4 \left\| V_I \right\|_1 , \tag{3}$$

where $V_I$ is constrained to be nonnegative, $V_I = \mathrm{Diag}(v_n)$, $V_I^{\max_k} = \mathrm{Diag}((v_n)^{\max_k})$, $C = C_1 C_2$, and $\lambda_2$, $\lambda_3$, and $\lambda_4$ are nonnegative regularization parameters.

More specifically, with $X^T \in \mathbb{R}^{m \times n}$, $V_I \in \mathbb{R}^{n \times n}$, $V_I^{\max_k} \in \mathbb{R}^{n \times n}$, $C \in \mathbb{R}^{n \times n}$, $C_1 \in \mathbb{R}^{n \times k}$, and $C_2 \in \mathbb{R}^{k \times n}$, we modify the traditional subspace learning for sample selection by introducing a sample scorer $V_I$, and a sample selector $V_I^{\max_k}$.

### 7.1 Experiment Setting for Core-subspace Learning

In experiments of this section, we set the number of epochs to be 200. We initialize the sample selection layer's weights by sampling from a uniform distribution $U[0.999999, 0.9999999]$ and the other layers' weights using the Xavier normal initializer. We use the Adam optimizer with a learning rate of 0.001 and set the hyperparameters $\lambda_2$, $\lambda_3$, and $\lambda_4$ in (3) to 2, 1, and 0.01, respectively.

For the number of selected samples $k$, we set it to 50, and we demonstrate our core-subspace learning procedure on MNIST-Fashion. We respectively use 1200, 1800, 2400, and 3000 randomly selected samples from original training samples to train, and 4000 randomly selected samples from original testing samples as the testing set. After training, we obtain the sample selector $V_I^{\max_k}$, i.e., the selected samples, then do sample reconstruction for testing samples based on these selected samples from training samples during testing time.

For evaluating our model, we measure reconstruction error $\eta$ in MSE.

### 7.2 Experimental Results

We compute the reconstruction error for testing samples after training the core-subspace learning model using different numbers of training samples 1200, 1800, 2400, and 3000. We plot the corresponding test errors in Figure 4. 1) It can be seen that the reconstruction error is small, which corroborates Assumption 2. 2) With the increase of the number of training samples, reconstruction error decreases. Such a phenomenon is due to the fact that more training samples make it more likely to observe training samples similar to the testing samples.

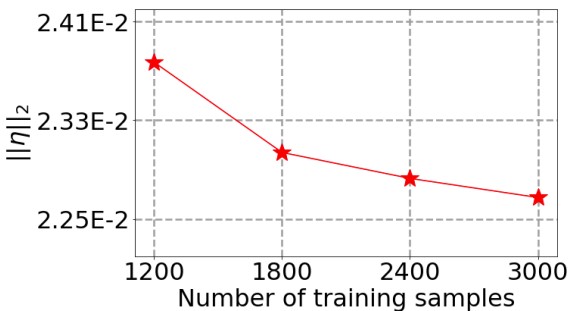

**Figure 4:** Reconstruction error for testing samples versus the number of training samples.

Furthermore, taking the case of 3000 training samples as an example, we visualize the sample selection from training samples and reconstruction of testing samples, as shown in Figures 5-7.

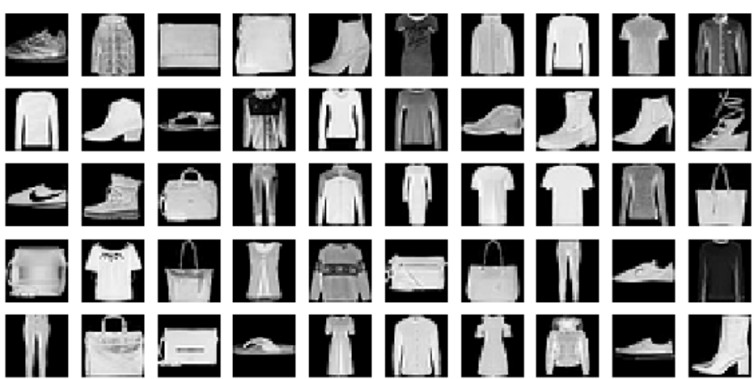

**Figure 5:** 50 selected samples from training samples.

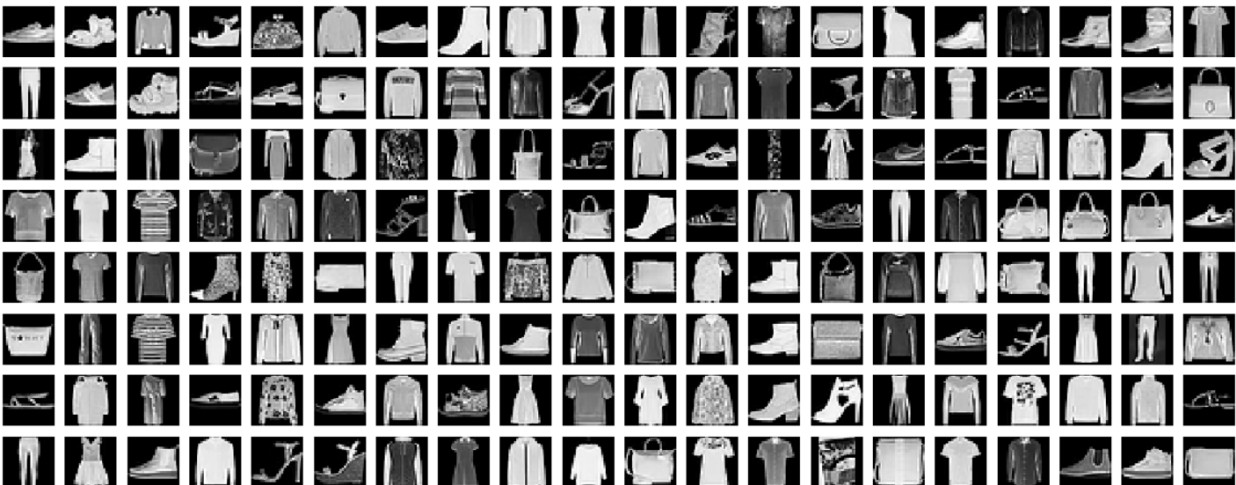

**Figure 6:** Testing samples. Here, we take 160 samples randomly for illustration.

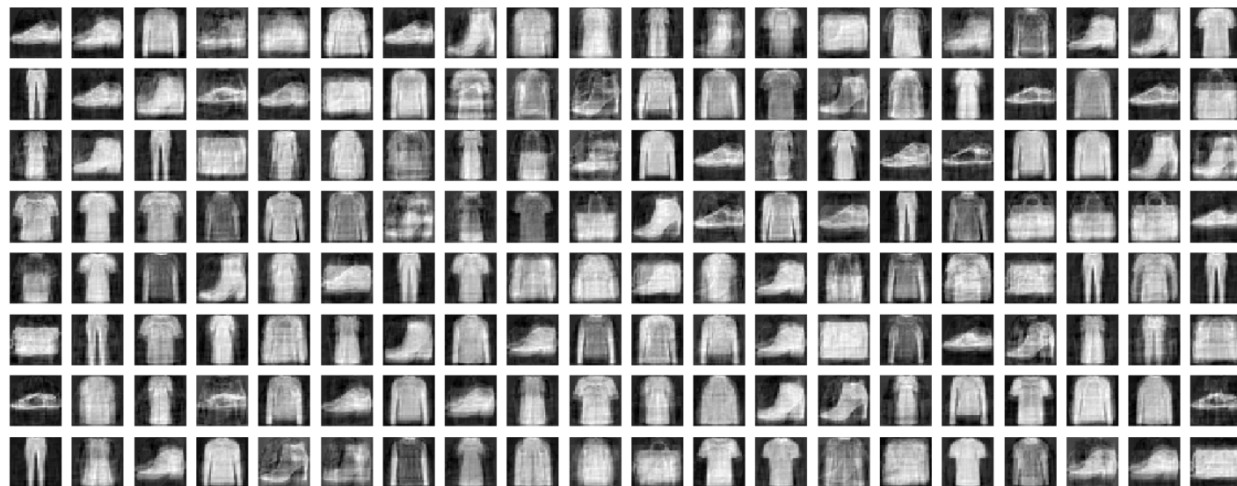

**Figure 7:** Reconstruction of testing samples based on 50 selected samples in Figure 5. Here, we illustrate the reconstructed images for those 160 testing samples given in Figure 6.

## 8 Detail of Assumption 3

Recall notations:

$$L_{emp}^{\text{score}}(\Phi, S) = \frac{1}{n} \sum_{i=1}^{n} \ell^{\text{score}}(\Phi, \mathrm{x}_i),$$

$$L_{emp}^{\text{selec}}(\Phi, S) = \frac{1}{n} \sum_{i=1}^{n} \ell^{\text{selec}}(\Phi, \mathrm{x}_i),$$

$$L_{emp}^{\text{score}, \backslash j}(\Phi, S) \triangleq \frac{1}{n} \sum_{i=1, i \neq j}^{n} \ell^{\text{score}}(\Phi, \mathrm{x}_i),$$

and

$$L_{emp}^{\text{selec}, \backslash j}(\Phi_*, S) \triangleq \frac{1}{n} \sum_{i=1, i \neq j}^{n} \ell^{\text{selec}}(\Phi, \mathrm{x}_i).$$

Without risking confusion, we usually omit $S$, and write $L_{emp}^{\text{score}}(\Phi, S)$, $L_{emp}^{\text{selec}}(\Phi, S)$, $L_{emp}^{\text{score}, \backslash j}(\Phi, S)$, and $L_{emp}^{\text{selec}, \backslash j}(\Phi)$ as $L_{emp}^{\text{score}}(\Phi)$, $L_{emp}^{\text{selec}}(\Phi)$, $L_{emp}^{\text{score}, \backslash j}(\Phi)$, and $L_{emp}^{\text{selec}, \backslash j}(\Phi)$, respectively.

About Assumption 3, a similar one is also adopted in [Le et al., 2018] (i.e., Assumption 6 in it), which requires that the difference between $L_{emp}^{\text{score}, \backslash j}(\cdot)$ and $L_{emp, \mathrm{z}}^{\text{score}}(\cdot)$ be small for the two $\Phi_*$ and $\Phi_*^{\backslash j}$. Assumption 3 only requires that the increase or decrease in error at the two points $\Phi_*^{\backslash j}$ and $\Phi_*$ be similar for $L_{emp}^{\text{score}, \backslash j}(\cdot)$ and $L_{emp, \mathrm{z}}^{\text{score}}(\cdot)$. Even if $L_{emp}^{\text{score}, \backslash j}(\cdot)$ is higher at $\Phi_*$ than $\Phi_*^{\backslash j}$, and $L_{emp, \mathrm{z}}^{\text{score}}(\Phi)$ is the opposite, the above bound can hold, because it simply requires that the difference of $L_{emp, \mathrm{z}}^{\text{score}}(\Phi)$ between $\Phi_*^{\backslash j}$ and $\Phi_*$ be bounded above by the difference of $L_{emp}^{\text{score}, \backslash j}(\cdot)$ between $\Phi_*$ and $\Phi_*^{\backslash j}$, up to some constant factor $n/(n-1)$. Additionally, note that $L_{emp, \mathrm{z}}^{\text{score}}(\Phi) = (n/(n-1))L_{emp}^{\text{score}, \backslash j}(\Phi)$ when $\mathcal{Z} = S^{\backslash j}$, so we have enhanced the factor $t$ into $n/(n-1)$ on the right-hand side of the equation in Assumption 3, which makes our Assumption 3 weaker and more reasonable than that in [Le et al., 2018].

For $0 \leqslant \Delta^t(L_{emp, \mathrm{z}}^{\text{score}}(\Phi_*), \Phi_*^{\backslash j}) + \Delta^t(L_{emp, \mathrm{z}}^{\text{score}}(\Phi_*^{\backslash j}), \Phi_*)$ below Assumption 3, the proof is as follows.

*Proof.* Note that $L_{emp, \mathrm{z}}^{\text{score}}(\cdot)$ is convex, so we have

$$
\begin{aligned}
& L_{emp, \mathrm{z}}^{\text{score}}((1-t)\Phi_* + t(\Phi_*^{\backslash j})) \\
\leqslant \; & (1-t)L_{emp, \mathrm{z}}^{\text{score}}(\Phi_*) + tL_{emp, \mathrm{z}}^{\text{score}}(\Phi_*^{\backslash j}),
\end{aligned}
\tag{4}
$$

and

$$
\begin{aligned}
& L_{emp, \mathrm{z}}^{\text{score}}((1-t)\Phi_*^{\backslash j} + t(\Phi_*)) \\
\leqslant \; & (1-t)L_{emp, \mathrm{z}}^{\text{score}}(\Phi_*^{\backslash j}) + tL_{emp, \mathrm{z}}^{\text{score}}(\Phi_*).
\end{aligned}
\tag{5}
$$

By (4) and (5), we have

$$
\begin{aligned}
& \Delta^t(L_{emp, \mathrm{z}}^{\text{score}}(\Phi_*), \Phi_*^{\backslash j}) + \Delta^t(L_{emp, \mathrm{z}}^{\text{score}}(\Phi_*^{\backslash j}), \Phi_*) \\
= \; & L_{emp, \mathrm{z}}^{\text{score}}(\Phi_*) - L_{emp, \mathrm{z}}^{\text{score}}((1-t)\Phi_* + t(\Phi_*^{\backslash j})) \\
& + L_{emp, \mathrm{z}}^{\text{score}}(\Phi_*^{\backslash j}) - L_{emp, \mathrm{z}}^{\text{score}}((1-t)\Phi_*^{\backslash j} + t(\Phi_*)) \\
\geqslant \; & L_{emp, \mathrm{z}}^{\text{score}}(\Phi_*) - (1-t)L_{emp, \mathrm{z}}^{\text{score}}(\Phi_*) - tL_{emp, \mathrm{z}}^{\text{score}}(\Phi_*^{\backslash j}) \\
& + L_{emp, \mathrm{z}}^{\text{score}}(\Phi_*^{\backslash j}) - (1-t)L_{emp, \mathrm{z}}^{\text{score}}(\Phi_*^{\backslash j}) - tL_{emp, \mathrm{z}}^{\text{score}}(\Phi_*) \\
= \; & (1-t)L_{emp, \mathrm{z}}^{\text{score}}(\Phi_*) - (1-t)L_{emp, \mathrm{z}}^{\text{score}}(\Phi_*) \\
& + (1-t)L_{emp, \mathrm{z}}^{\text{score}}(\Phi_*^{\backslash j}) - (1-t)L_{emp, \mathrm{z}}^{\text{score}}(\Phi_*^{\backslash j}) \\
= \; & 0.
\end{aligned}
$$

$\square$

## 9 Proofs of Theorems 1 and 2

Before giving the proofs of Theorems 1 and 2, firstly, we list some known definitions, properties, and two theorems; these materials can be found in the literature and are given here for the preparation to prove our new theorems. Then, we introduce five lemmas, all of which will be used to prove our new Theorems 1 and 2.

**Definition A** ($\sigma$-Admissibility). *If a loss function $\ell$ defined on $\mathcal{X} \times \mathcal{X}$ is convex with respect to its first argument, $\exists \sigma \in (0, +\infty)$ such that $\forall \mathrm{x}_1, \mathrm{x}_2, \mathrm{x}_3 \in \mathcal{X}$, and $\forall f \in \mathcal{H}$, the following inequality holds:*

$$|\ell(f(\mathrm{x}_1), \mathrm{x}_3) - \ell(f(\mathrm{x}_2), \mathrm{x}_3)| \leqslant \sigma \|f(x_1) - f(x_2)\|_2,$$

*then we say that $\ell$ is $\sigma$-admissible with respect to $\mathcal{H}$.*

**Definition B** (Bregman Divergence [Mohri et al., 2018]). *Let $F : \mathcal{H} \to \mathbb{R}$ be a convex function. $\forall f, g \in \mathcal{H}$,*

$$B_F(f\|g) = F(f) - F(g) - \langle f - g, \nabla F(g) \rangle,$$

*where $\nabla F(g)$ is the subgradient of $F$ at $g$. Bregman divergence $B_F(f\|g)$ measures the difference of $F(f)$ and its linear approximation.*

**Definition C** (Convexity [Nesterov, 2018]). *A function $f : \mathbb{R}^m \to \mathbb{R}$ is convex if its domain is a convex set and for any $x$ and $y$ in its domain, and $\forall t \in [0, 1]$,*

$$f(tx + (1 - t)y) \leqslant tf(x) + (1 - t)f(y).$$

**Definition D** ($c$-Strongly Convex [Nesterov, 2018]). *A function $f : \mathbb{R}^m \to \mathbb{R}$ is $c$-strongly convex if its domain is a convex set and for any $x$ and $y$ in its domain, the following inequality holds:*

$$\langle x - y, \nabla f(x) - \nabla f(y) \rangle \geqslant c \|x - y\|_2^2.$$

**Property A** ([Mohri et al., 2018]). *For a Bregman divergence $B_F(f\|g)$, where $F : \mathcal{H} \to \mathbb{R}$, and $f$ and $g \in \mathcal{H}$, we have*

$$B_F(f\|g) \geqslant 0.$$

**Theorem A** ( [Bousquet and Elisseeff, 2002]). *Let $\boldsymbol{A}$ has uniform stability $\beta$ with respect to the loss function $\ell$ such that $\ell(\boldsymbol{A}_S, \mathbf{x}) \leqslant \kappa_5$, for all $\mathbf{x} \in \mathcal{X}$ and all sets $S$. Then, for any $n \geqslant 1$ and any $\delta \in (0, 1)$, with probability at least $1 - \delta$,*

$$L(\boldsymbol{A}, S) \leqslant L_{emp}(\boldsymbol{A}, S) + 2\beta + (4n\beta + \kappa_5)\sqrt{\frac{\ln\left(\frac{1}{\delta}\right)}{2n}}, \tag{6}$$

*and*

$$L(\boldsymbol{A}, S) \leqslant L_{loo}(\boldsymbol{A}, S) + \beta + (4n\beta + \kappa_5)\sqrt{\frac{\ln\left(\frac{1}{\delta}\right)}{2n}}. \tag{7}$$

**Lemma A.** $\ell^{\mathrm{selec}}$ *and* $\ell^{\mathrm{score}}$ *are both $\sigma$-admissible, and $\sigma^{\mathrm{selec}}$ and $\sigma^{\mathrm{score}}$ are respectively equal to*

$$\kappa_1 \left\| \left( \Phi_*^{\max_k} + \left( \Phi_*^{\backslash j} \right)^{\max_k} \right) \mathrm{W_E W_D} \right\|_2 + 2\kappa_1,$$

*and*

$$\kappa_1 \left\| \left( \Phi_* + \Phi_*^{\backslash j} \right) \mathrm{W_E W_D} \right\|_2 + 2\kappa_1.$$

*Proof.* $\forall \mathrm{x} \in \mathcal{X}$,

$$\left| \ell^{\mathrm{selec}} \left( \Phi_*^{\max_k}, \mathrm{x} \right) - \ell^{\mathrm{selec}} \left( \left( \Phi_*^{\backslash j} \right)^{\max_k}, \mathrm{x} \right) \right|$$

$$= \left| \left\| \mathrm{x} - \left( (\mathrm{x} \Phi_*^{\max_k}) \mathrm{W_E} \right) \mathrm{W_D} \right\|_2^2 - \left\| \mathrm{x} - \left( \left( \mathrm{x} \left( \Phi_*^{\backslash j} \right)^{\max_k} \right) \mathrm{W_E} \right) \mathrm{W_D} \right\|_2^2 \right|$$

$$= \left\langle \left( \left( \mathrm{x} \left( \Phi_*^{\max_k} - \left( \Phi_*^{\backslash j} \right)^{\max_k} \right) \right) \mathrm{W_E} \right) \mathrm{W_D}, \right.$$

$$\left. \left( (\mathrm{x} \Phi_*^{\max_k}) \mathrm{W_E} \right) \mathrm{W_D} + \left( \left( \mathrm{x} \left( \Phi_*^{\backslash j} \right)^{\max_k} \right) \mathrm{W_E} \right) \mathrm{W_D} - 2\mathrm{x} \right\rangle \qquad (8)$$

$$\leqslant \left\| \left( \left( \mathrm{x} \left( \Phi_*^{\max_k} - \left( \Phi_*^{\backslash j} \right)^{\max_k} \right) \right) \mathrm{W_E} \right) \mathrm{W_D} \right\|_2 \cdot$$

$$\left\| \left( (\mathrm{x} \Phi_*^{\max_k}) \mathrm{W_E} \right) \mathrm{W_D} + \left( \left( \mathrm{x} \left( \Phi_*^{\backslash j} \right)^{\max_k} \right) \mathrm{W_E} \right) \mathrm{W_D} - 2\mathrm{x} \right\|_2$$

$$\leqslant \left( \|\mathrm{x}\|_2 \left\| \left( \Phi_*^{\max_k} + \left( \Phi_*^{\backslash j} \right)^{\max_k} \right) \mathrm{W_E W_D} \right\|_2 + 2 \|\mathrm{x}\|_2 \right) \cdot$$

$$\left\| \left( \left( \mathrm{x} \left( \Phi_*^{\max_k} - \left( \Phi_*^{\backslash j} \right)^{\max_k} \right) \right) \mathrm{W_E} \right) \mathrm{W_D} \right\|_2 .$$

Similarly, we have

$$\left| \ell^{\mathrm{score}} \left( \Phi_*, \mathrm{x} \right) - \ell^{\mathrm{score}} \left( \Phi_*^{\backslash j}, \mathrm{x} \right) \right|$$

$$\leqslant \left( \|\mathrm{x}\|_2 \left\| \left( \Phi_* + \Phi_*^{\backslash j} \right) \mathrm{W_E W_D} \right\|_2 + 2 \|\mathrm{x}\|_2 \right) \left\| \left( \left( \mathrm{x} \left( \Phi_* - \Phi_*^{\backslash j} \right) \right) \mathrm{W_E} \right) \mathrm{W_D} \right\|_2 .$$

$\square$

**Lemma B.**

$$\sigma^{\mathrm{selec}} \leqslant \sigma^{\mathrm{score}}.$$

*Proof.* Let

$$\Phi_*^{\max_k} + \left( \Phi_*^{\backslash j} \right)^{\max_k} = \mathrm{W_0} \left( \Phi_* + \Phi_*^{\backslash j} \right),$$

where the entries of $\mathrm{W_0}$ are zeros except $k$ ones on its diagonal.

Then, we have

$$\left\| \left( \Phi_*^{\max_k} + \left( \Phi_*^{\backslash j} \right)^{\max_k} \right) \mathrm{W_E W_D} \right\|_2$$

$$= \left\| \mathrm{W_0} \left( \Phi_* + \Phi_*^{\backslash j} \right) \mathrm{W_E W_D} \right\|_2$$

$$\leqslant \|\mathrm{W_0}\|_2 \left\| \left( \Phi_* + \Phi_*^{\backslash j} \right) \mathrm{W_E W_D} \right\|_2$$

$$= \left\| \left( \Phi_* + \Phi_*^{\backslash j} \right) \mathrm{W_E W_D} \right\|_2 .$$

So, we have proved Lemma B. $\square$

**Lemma C.** $\ell^{\mathrm{score}}$ *and* $\ell^{\mathrm{selec}}$ *are 2-strongly convex with respect to* $((\mathrm{x}(\Phi(\mathrm{W_I})^{\max_k}))\mathrm{W_E})\mathrm{W_D}$ *and* $((\mathrm{x}\Phi(\mathrm{W_I}))\mathrm{W_E})\mathrm{W_D}$, *respectively.*

*Proof.* It follows by using Definition D. $\qquad\square$

**Lemma D.**

$$\lambda_1 B_{L_{emp}^{\mathrm{score},\backslash j}}\left(\Phi \| \Phi^{\backslash j}\right) \leqslant B_{RL_{emp}^{\backslash j}}\left(\Phi \| \Phi^{\backslash j}\right), \tag{9}$$

$$\lambda_1 B_{L_{emp}^{\mathrm{score},\backslash j}}\left(\Phi^{\backslash j} \| \Phi\right) \leqslant B_{RL_{emp}}\left(\Phi^{\backslash j} \| \Phi\right), \tag{10}$$

*and*

$$\lambda_1 B_{L_{emp}^{\mathrm{score}}}\left(\Phi^{\backslash j} \| \Phi\right) \leqslant B_{RL_{emp}}\left(\Phi^{\backslash j} \| \Phi\right). \tag{11}$$

*Proof.* Let $L_{emp}^{\mathrm{selec},\backslash j}(\Phi) = 1/n \sum_{i=1, i \neq j}^{n} \ell^{\mathrm{selec}}(\Phi, \mathrm{x}_i)$. Note that $\ell^{\mathrm{selec}}(\Phi, \mathrm{x}_i)$ is convex with respect to $\Phi$, and so is $L_{emp}^{\mathrm{selec},\backslash j}(\Phi)$. By Property A, we have

$$0 \leqslant B_{RL_{emp}^{\backslash j}}\left(\Phi \| \Phi^{\backslash j}\right) - \lambda_1 B_{L_{emp}^{\mathrm{score},\backslash j}}\left(\Phi \| \Phi^{\backslash j}\right).$$

Then, (9) follows.

Similarly, by the convexity of $L_{emp}^{\mathrm{selec},\backslash j}(\Phi)$ and $\ell^{\mathrm{score}}(\Phi, \mathrm{x}_i)$, we have (10) and (11). $\qquad\square$

**Lemma E.** *If Assumption 3 holds, then we have,* $\forall t \in [0, 1]$,

$$B_{L_{emp,\mathrm{z}}^{\mathrm{score}}}\left(\Phi_*^{\backslash j} \| \Phi_*\right) + B_{L_{emp,\mathrm{z}}^{\mathrm{score}}}\left(\Phi_* \| \Phi_*^{\backslash j}\right)$$

$$\leqslant \quad \left(\frac{n}{n-1}\right)\left(B_{L_{emp}^{\mathrm{score},\backslash j}}\left(\Phi_*^{\backslash j} \| \Phi_*\right) + B_{L_{emp}^{\mathrm{score},\backslash j}}\left(\Phi_* \| \Phi_*^{\backslash j}\right)\right).$$

*Proof.* By Definition B and the linearity of the inner product, we have

$$B_{L_{emp,\mathrm{z}}^{\mathrm{score}}}\left(\Phi_*^{\backslash j} \| \Phi_*\right) + B_{L_{emp,\mathrm{z}}^{\mathrm{score}}}\left(\Phi_* \| \Phi_*^{\backslash j}\right)$$

$$\leqslant \quad L_{emp,\mathrm{z}}^{\mathrm{score}}\left(\Phi_*^{\backslash j}\right) - L_{emp,\mathrm{z}}^{\mathrm{score}}\left(\Phi_*\right) - \left\langle \Phi_*^{\backslash j} - \Phi_*, \nabla L_{emp,\mathrm{z}}^{\mathrm{score}}\left(\Phi_*\right)\right\rangle$$

$$+ L_{emp,\mathrm{z}}^{\mathrm{score}}\left(\Phi_*\right) - L_{emp,\mathrm{z}}^{\mathrm{score}}\left(\Phi_*^{\backslash j}\right) - \left\langle \Phi_* - \Phi_*^{\backslash j}, \nabla L_{emp,\mathrm{z}}^{\mathrm{score}}\left(\Phi_*^{\backslash j}\right)\right\rangle$$

$$= \quad -\left\langle \Phi_*^{\backslash j} - \Phi_*, \nabla L_{emp,\mathrm{z}}^{\mathrm{score}}\left(\Phi_*\right)\right\rangle - \left\langle \Phi_* - \Phi_*^{\backslash j}, \nabla L_{emp,\mathrm{z}}^{\mathrm{score}}\left(\Phi_*^{\backslash j}\right)\right\rangle$$

$$= \quad \lim_{t \to 0^+}\left(\frac{L_{emp,\mathrm{z}}^{\mathrm{score}}(\Phi_*) - L_{emp,\mathrm{z}}^{\mathrm{score}}((1-t)\Phi_* + t(\Phi_*^{\backslash j}))}{t}\right)$$

$$+ \lim_{t \to 0^+}\left(\frac{L_{emp,\mathrm{z}}^{\mathrm{score}}(\Phi_*^{\backslash j}) - L_{emp,\mathrm{z}}^{\mathrm{score}}((1-t)\Phi_*^{\backslash j} + t(\Phi_*))}{t}\right)$$

$$\leqslant \quad \left(\frac{n}{n-1}\right)\lim_{t \to 0^+}\left(\frac{L_{emp}^{\mathrm{score},\backslash j}(\Phi_*) - L_{emp}^{\mathrm{score},\backslash j}((1-t)\Phi_* + t(\Phi_*^{\backslash j}))}{t}\right)$$

$$+ \left(\frac{n}{n-1}\right)\lim_{t \to 0^+}\left(\frac{L_{emp}^{\mathrm{score},\backslash j}(\Phi_*^{\backslash j}) - L_{emp}^{\mathrm{score},\backslash j}((1-t)\Phi_*^{\backslash j} + t(\Phi_*))}{t}\right)$$

$$= \quad \left(\frac{n}{n-1}\right)\left(B_{L_{emp}^{\mathrm{score},\backslash j}}\left(\Phi_*^{\backslash j} \| \Phi_*\right) + B_{L_{emp}^{\mathrm{score},\backslash j}}\left(\Phi_* \| \Phi_*^{\backslash j}\right)\right),$$

where the third equation from the bottom is obtained by the definition of directional derivatives, and the penultimate inequality is derived by Assumption 3 of the main text. $\square$

### 9.1 Proof of Theorem 1

Let

$$RL_{emp}(\Phi, S) \triangleq \frac{1}{n} \left( \sum_{x_i \in S} \ell^{\text{selec}}(\Phi^{\max_k}, x_i) + \lambda_1 \ell^{\text{score}}(\Phi, x_i) \right), \tag{12}$$

and

$$RL_{emp}^{\backslash j}(\Phi, S) \triangleq \frac{1}{n} \left( \sum_{x_i \in S^{\backslash j}} \ell^{\text{selec}}(\Phi^{\max_k}, x_i) + \lambda_1 \ell^{\text{score}}(\Phi, x_i) \right), \tag{13}$$

where $j \in \{1, \dots, n\}$.

**Theorem 1** (Uniform Stability). *Under Assumptions 1, 2, and 3, we have, $\forall n \geqslant 2$,*

$$\left\| \ell^{\text{selec}} \left( \Phi_*^{\max_k}, \cdot \right) - \ell^{\text{selec}} \left( \left( \Phi_*^{\backslash j} \right)^{\max_k}, \cdot \right) \right\|_{\infty}$$

$$\leqslant \frac{\sigma^{\text{selec}} \sigma^{\text{score}} \kappa_2 \kappa_4^2 \sqrt{u(\kappa_2 + \lambda_1)} \sqrt{u(\kappa_2 + \lambda_1) + 8\lambda_1 \kappa_3 \kappa_5} + u \sigma^{\text{selec}} \sigma^{\text{score}} \kappa_2 \kappa_4^2 (\kappa_2 + \lambda_1)}{4\lambda_1 (n-1)} \tag{14}$$

$$+ \frac{\sigma^{\text{selec}} \kappa_2 \kappa_3 \kappa_5}{(n-1)},$$

*where $\kappa_5 = \|((\Phi_* - \Phi_*^{\backslash j})W_E)W_D\|_2$, $\sigma^{\text{selec}} = \kappa_1 \|(\Phi_*^{\max_k} + (\Phi_*^{\backslash j})^{\max_k})W_E W_D\|_2 + 2\kappa_1$, and $\sigma^{\text{score}} = \kappa_1 \|(\Phi_* + \Phi_*^{\backslash j})W_E W_D\|_2 + 2\kappa_1$.*

*Further, the convergence rate of uniform stability bound in $n$ and $\lambda_1$ is*

$$\left\| \ell^{\text{selec}} \left( \Phi_*^{\max_k}, \cdot \right) - \ell^{\text{selec}} \left( \left( \Phi_*^{\backslash j} \right)^{\max_k}, \cdot \right) \right\|_{\infty} = \mathcal{O} \left( \frac{1}{n \min\{\sqrt{\lambda_1}, \lambda_1\}} + \frac{1}{n} \right).$$

*Proof.* By Lemmas D and E, and Definition B, we have

$$\left( \frac{n}{n-1} \right) \left( B_{RL_{emp}^{\backslash j}} \left( \Phi_* \| \Phi_*^{\backslash j} \right) + B_{RL_{emp}} \left( \Phi_*^{\backslash j} \| \Phi_* \right) \right)$$

$$\geqslant \lambda_1 \left( \frac{n}{n-1} \right) \left( B_{L_{emp}^{\text{score},\backslash j}} \left( \Phi_* \| \Phi_*^{\backslash j} \right) + B_{L_{emp}^{\text{score},\backslash j}} \left( \Phi_*^{\backslash j} \| \Phi_* \right) \right)$$

$$\geqslant \lambda_1 B_{L_{emp,z}^{\text{score}}} \left( \Phi_*^{\backslash j} \| \Phi_* \right) + B_{L_{emp,z}^{\text{score}}} \left( \Phi_* \| \Phi_*^{\backslash j} \right)$$

$$= \frac{\lambda_1}{u} \sum_{i=1}^{u} \left\langle z_i \left( \Phi_*^{\backslash j} - \Phi_* \right) W_E W_D, \nabla L_{emp}^{\text{score},\backslash j}(\Phi_*^{\backslash j}) - \nabla L_{emp}^{\text{score},\backslash j}(\Phi_*) \right\rangle.$$

By Lemma C, we obtain,

$$\left( \frac{n}{n-1} \right) \left( B_{RL_{emp}^{\backslash j}} \left( \Phi_* \| \Phi_*^{\backslash j} \right) + B_{RL_{emp}} \left( \Phi_*^{\backslash j} \| \Phi_* \right) \right)$$

$$\geqslant \frac{2\lambda_1}{u} \sum_{i=1}^{u} \left\| z_i \left( \Phi_*^{\backslash j} - \Phi_* \right) W_E W_D \right\|_2^2. \tag{15}$$

Because $(\Phi_*^{\max_k}, \Phi_*)$ and $((\Phi_*^{\backslash j})^{\max_k}, (\Phi_*^{\backslash j}))$ are the optimal feature selectors and feature scorers for the errors (12) and (13), respectively, we have

$$
\begin{aligned}
& B_{RL_{emp}^{\backslash j}}\left(\Phi_* \| \Phi_*^{\backslash j}\right) + B_{RL_{emp}}\left(\Phi_*^{\backslash j} \| \Phi_*\right) \\
=\; & RL_{emp}^{\backslash j}\left(\Phi_*, S\right) - RL_{emp}^{\backslash j}\left(\Phi_*^{\backslash j}, S\right) + RL_{emp}\left(\Phi_*^{\backslash j}, S\right) - RL_{emp}\left(\Phi_*, S\right) \\
=\; & RL_{emp}^{\backslash j}\left(\Phi_*, S\right) - RL_{emp}\left(\Phi_*, S\right) + RL_{emp}\left(\Phi_*^{\backslash j}, S\right) - RL_{emp}^{\backslash j}\left(\Phi_*^{\backslash j}, S\right) \\
=\; & -\frac{1}{n}\ell^{\mathrm{selec}}\left(\Phi_*^{\max_k}, \mathrm{x}_j\right) - \frac{1}{n}\lambda_1\ell^{\mathrm{score}}\left(\Phi_*, \mathrm{x}_j\right) + \frac{1}{n}\ell^{\mathrm{selec}}\left(\left(\Phi_*^{\backslash j}\right)^{\max_k}, \mathrm{x}_j\right) \\
& + \frac{1}{n}\lambda_1\ell^{\mathrm{score}}\left(\Phi_*^{\backslash j}, \mathrm{x}_j\right) \\
=\; & \frac{1}{n}\ell^{\mathrm{selec}}\left(\left(\Phi_*^{\backslash j}\right)^{\max_k}, \mathrm{x}_j\right) - \frac{1}{n}\ell^{\mathrm{selec}}\left(\Phi_*^{\max_k}, \mathrm{x}_j\right) \\
& + \frac{1}{n}\lambda_1\ell^{\mathrm{score}}\left(\Phi_*^{\backslash j}, \mathrm{x}_j\right) - \frac{1}{n}\lambda_1\ell^{\mathrm{score}}\left(\Phi_*, \mathrm{x}_j\right).
\end{aligned}
\tag{16}
$$

Plugging (16) into (15), we get

$$
\begin{aligned}
& \frac{2\lambda_1}{u}\sum_{i=1}^{u}\left\|\mathrm{z}_i\left(\Phi_*^{\backslash j} - \Phi_*\right)\mathrm{W_E}\mathrm{W_D}\right\|_2^2 \\
\leqslant\; & \left(\frac{n}{n-1}\right)\left(\frac{1}{n}\ell^{\mathrm{selec}}\left(\left(\Phi_*^{\backslash j}\right)^{\max_k}, \mathrm{x}_j\right) - \frac{1}{n}\ell^{\mathrm{selec}}\left(\Phi_*^{\max_k}, \mathrm{x}_j\right)\right) \\
& + \left(\frac{n}{n-1}\right)\left(\frac{1}{n}\lambda_1\ell^{\mathrm{score}}\left(\Phi_*^{\backslash j}, \mathrm{x}_j\right) - \frac{1}{n}\lambda_1\ell^{\mathrm{score}}\left(\Phi_*, \mathrm{x}_j\right)\right).
\end{aligned}
$$

By Lemma A and Assumption 1, we have

$$
\begin{aligned}
& \frac{2\lambda_1}{u}\sum_{i=1}^{u}\left\|\mathrm{z}_i\left(\Phi_*^{\backslash j} - \Phi_*\right)\mathrm{W_E}\mathrm{W_D}\right\|_2^2 \\
\leqslant\; & \left(\frac{n}{n-1}\right)\frac{\sigma^{\mathrm{selec}}\left\|\left(\mathrm{x}_j\left(\Phi_*^{\max_k} - \left(\Phi_*^{\backslash j}\right)^{\max_k}\right)\mathrm{W_E}\right)\mathrm{W_D}\right\|_2}{n} \\
& + \left(\frac{n}{n-1}\right)\frac{\lambda_1\sigma^{\mathrm{score}}\left\|\left(\left(\mathrm{x}_j\left(\Phi_* - \Phi_*^{\backslash j}\right)\right)\mathrm{W_E}\right)\mathrm{W_D}\right\|_2}{n} \\
\leqslant\; & \left(\frac{n}{n-1}\right)\frac{\sigma^{\mathrm{selec}}\kappa_2\left\|\left(\mathrm{x}_j\left(\Phi_* - \Phi_*^{\backslash j}\right)\mathrm{W_E}\right)\mathrm{W_D}\right\|_2}{n} \\
& + \left(\frac{n}{n-1}\right)\frac{\lambda_1\sigma^{\mathrm{score}}\left\|\left(\left(\mathrm{x}_j\left(\Phi_* - \Phi_*^{\backslash j}\right)\right)\mathrm{W_E}\right)\mathrm{W_D}\right\|_2}{n}.
\end{aligned}
\tag{17}
$$

Let $\kappa_5 = \left\| \left( \left( \Phi_* - \Phi_*^{\backslash j} \right) W_E \right) W_D \right\|_2$. Based on Assumption 2, $\forall x \in \mathcal{X}$, we have

$$
\begin{aligned}
& \left\| \left( x \left( \Phi_* - \Phi_*^{\backslash j} \right) W_E \right) W_D \right\|_2 \\
= \; & \left\| \sum_{i=1}^{u} \left( \alpha_i z_i \left( \Phi_* - \Phi_*^{\backslash j} \right) W_E \right) W_D + \left( \eta \left( \Phi_* - \Phi_*^{\backslash j} \right) W_E \right) W_D \right\|_2 \\
\leqslant \; & \sqrt{\sum_{i=1}^{u} \alpha_i^2} \sqrt{\sum_{i=1}^{u} \left\| \left( z_i \left( \Phi_* - \Phi_*^{\backslash j} \right) W_E \right) W_D \right\|_2^2} + \|\eta\|_2 \left\| \left( \left( \Phi_* - \Phi_*^{\backslash j} \right) W_E \right) W_D \right\|_2 \\
\leqslant \; & \kappa_4 \sqrt{\sum_{i=1}^{u} \left\| \left( z_i \left( \Phi_* - \Phi_*^{\backslash j} \right) W_E \right) W_D \right\|_2^2} + \frac{\kappa_3 \kappa_5}{n}.
\end{aligned}
\tag{18}
$$

Combining (17) and (18), we get

$$
\begin{aligned}
& \frac{2\lambda_1}{u} \sum_{i=1}^{u} \left\| z_i \left( \Phi_*^{\backslash j} - \Phi_* \right) W_E W_D \right\|_2^2 \\
\leqslant \; & \left( \frac{1}{n-1} \right) \left( \sigma^{\text{selec}} \kappa_2 \kappa_4 \sqrt{\sum_{i=1}^{u} \left\| \left( z_i \left( \Phi_* - \Phi_*^{\backslash j} \right) W_E \right) W_D \right\|_2^2} \right. \\
& \left. + \lambda_1 \sigma^{\text{score}} \kappa_4 \sqrt{\sum_{i=1}^{u} \left\| \left( z_i \left( \Phi_* - \Phi_*^{\backslash j} \right) W_E \right) W_D \right\|_2^2} + \frac{\sigma^{\text{selec}} \kappa_2 \kappa_3 \kappa_5}{n} + \frac{\lambda_1 \sigma^{\text{score}} \kappa_3 \kappa_5}{n} \right).
\end{aligned}
\tag{19}
$$

From (19), we obtain

$$
\begin{aligned}
& \sqrt{\sum_{i=1}^{u} \left\| \left( z_i \left( \Phi_* - \Phi_*^{\backslash j} \right) W_E \right) W_D \right\|_2^2} \\
\leqslant \; & \sqrt{\left( \frac{u \left( \sigma^{\text{selec}} \kappa_2 \kappa_4 + \lambda_1 \sigma^{\text{score}} \kappa_4 \right)}{4\lambda_1 (n-1)} \right)^2 + \frac{u \left( \sigma^{\text{selec}} \kappa_2 \kappa_3 \kappa_5 + \lambda_1 \sigma^{\text{score}} \kappa_3 \kappa_5 \right)}{2\lambda_1 n (n-1)}} \\
& + \frac{u \left( \sigma^{\text{selec}} \kappa_2 \kappa_4 + \lambda_1 \sigma^{\text{score}} \kappa_4 \right)}{4\lambda_1 (n-1)}.
\end{aligned}
\tag{20}
$$

Plugging (20) into (18), we have

$$
\begin{aligned}
& \left\| \left( x \left( \Phi_* - \Phi_*^{\backslash j} \right) W_E \right) W_D \right\|_2 \\
\leqslant \; & \kappa_4 \sqrt{\left( \frac{u \left( \sigma^{\text{selec}} \kappa_2 \kappa_4 + \lambda_1 \sigma^{\text{score}} \kappa_4 \right)}{4\lambda_1 (n-1)} \right)^2 + \frac{u \left( \sigma^{\text{selec}} \kappa_2 \kappa_3 \kappa_5 + \lambda_1 \sigma^{\text{score}} \kappa_3 \kappa_5 \right)}{2\lambda_1 n (n-1)}} \\
& + \frac{u \kappa_4 \left( \sigma^{\text{selec}} \kappa_2 \kappa_4 + \lambda_1 \sigma^{\text{score}} \kappa_4 \right)}{4\lambda_1 (n-1)} + \frac{\kappa_3 \kappa_5}{n}.
\end{aligned}
\tag{21}
$$

And by Assumption 1, we get

$$\left| \ell^{\mathrm{selec}}\left(\Phi_*^{\max_k}, \mathrm{x}\right) - \ell^{\mathrm{selec}}\left(\left(\Phi_*^{\backslash j}\right)^{\max_k}, \mathrm{x}\right) \right|$$

$$\leqslant \quad \sigma^{\mathrm{selec}} \left\| \left( \left( \mathrm{x}\left(\Phi_*^{\max_k} - \left(\Phi_*^{\backslash j}\right)^{\max_k}\right)\right) \mathrm{W_E}\right) \mathrm{W_D} \right\|_2$$

$$\leqslant \quad \sigma^{\mathrm{selec}} \kappa_2 \kappa_4 \sqrt{\left(\frac{u\left(\sigma^{\mathrm{selec}}\kappa_2\kappa_4 + \lambda_1 \sigma^{\mathrm{score}}\kappa_4\right)}{4\lambda_1(n-1)}\right)^2 + \frac{u\left(\sigma^{\mathrm{selec}}\kappa_2\kappa_3\kappa_5 + \lambda_1 \sigma^{\mathrm{score}}\kappa_3\kappa_5\right)}{2\lambda_1 n(n-1)}}$$

$$+ \frac{u\sigma^{\mathrm{selec}}\kappa_2\kappa_4\left(\sigma^{\mathrm{selec}}\kappa_2\kappa_4 + \lambda_1\sigma^{\mathrm{score}}\kappa_4\right)}{4\lambda_1(n-1)} + \frac{\kappa_3\sigma^{\mathrm{selec}}\kappa_2\kappa_5}{n}$$

$$\leqslant \quad \frac{\sigma^{\mathrm{selec}}\kappa_2\kappa_4}{4\lambda_1(n-1)}\sqrt{u\left(\sigma^{\mathrm{selec}}\kappa_2 + \lambda_1\sigma^{\mathrm{score}}\right)}\sqrt{u\kappa_4^2\left(\sigma^{\mathrm{selec}}\kappa_2 + \lambda_1\sigma^{\mathrm{score}}\right) + 8\lambda_1\kappa_3\kappa_5}$$

$$+ \frac{u\sigma^{\mathrm{selec}}\kappa_2\kappa_4^2\left(\sigma^{\mathrm{selec}}\kappa_2 + \lambda_1\sigma^{\mathrm{score}}\right) + 4\lambda_1\sigma^{\mathrm{selec}}\kappa_2\kappa_3\kappa_5}{4\lambda_1(n-1)}.$$

Finally, by Lemma B, we have

$$\left| \ell^{\mathrm{selec}}\left(\Phi_*^{\max_k}, \mathrm{x}\right) - \ell^{\mathrm{selec}}\left(\left(\Phi_*^{\backslash j}\right)^{\max_k}, \mathrm{x}\right) \right|$$

$$\leqslant \quad \frac{\sigma^{\mathrm{selec}}\sigma^{\mathrm{score}}\kappa_2\kappa_4\sqrt{u\left(\kappa_2 + \lambda_1\right)}\sqrt{u\kappa_4^2\left(\kappa_2 + \lambda_1\right) + 8\lambda_1\kappa_3\kappa_5}}{4\lambda_1(n-1)}$$

$$+ \frac{u\sigma^{\mathrm{selec}}\sigma^{\mathrm{score}}\kappa_2\kappa_4^2\left(\kappa_2 + \lambda_1\right) + 4\lambda_1\sigma^{\mathrm{selec}}\kappa_2\kappa_3\kappa_5}{4\lambda_1(n-1)}$$

$$\leqslant \quad \frac{\sigma^{\mathrm{selec}}\sigma^{\mathrm{score}}\kappa_2\kappa_4^2\sqrt{u\left(\kappa_2 + \lambda_1\right)}\sqrt{u\left(\kappa_2 + \lambda_1\right) + 8\lambda_1\kappa_3\kappa_5} + u\sigma^{\mathrm{selec}}\sigma^{\mathrm{score}}\kappa_2\kappa_4^2\left(\kappa_2 + \lambda_1\right)}{4\lambda_1(n-1)}$$

$$+ \frac{\sigma^{\mathrm{selec}}\kappa_2\kappa_3\kappa_5}{(n-1)}.$$

$\square$

## 10.2 Proof of Theorem 2

**Theorem 2** (Generalization Error). [5] $\exists \kappa_5 > 0$ and $\delta \in (0,1)$, $\forall \mathrm{x} \in \mathcal{X}$ and $S$, as long as $\ell(\boldsymbol{A}_S, \mathrm{x}) \leqslant \kappa_5$, the following inequality holds with probability at least $1 - \delta$,

$$L^{\mathrm{selec}}\left(\Phi_*^{\max_k}, S\right) - L_{emp}^{\mathrm{selec}}\left(\Phi_*^{\max_k}, S\right)$$

$$\leqslant \frac{\sigma^{\mathrm{selec}}\sigma^{\mathrm{score}}\kappa_2\kappa_4^2\left(1 + \sqrt{2n\ln\left(\frac{1}{\delta}\right)}\right)\sqrt{u\left(\kappa_2 + \lambda_1\right)}\sqrt{u\left(\kappa_2 + \lambda_1\right) + 8\lambda_1\kappa_3\kappa_5}}{2\lambda_1(n-1)} + \kappa_5\sqrt{\frac{\ln\left(\frac{1}{\delta}\right)}{2n}} \quad (22)$$

$$+ \left(1 + \sqrt{2n\ln\left(\frac{1}{\delta}\right)}\right)\left(\frac{u\sigma^{\mathrm{selec}}\sigma^{\mathrm{score}}\kappa_2\kappa_4^2\left(\kappa_2 + \lambda_1\right) + 4\lambda_1\sigma^{\mathrm{selec}}\kappa_2\kappa_3\kappa_5}{2\lambda_1(n-1)}\right),$$

and

$$L^{\mathrm{selec}}\left(\Phi_*^{\max_k}, S\right) - L_{loo}^{\mathrm{selec}}\left(\Phi_*^{\max_k}, S\right)$$

$$\leqslant \frac{\sigma^{\mathrm{selec}}\sigma^{\mathrm{score}}\kappa_2\kappa_4^2\left(1 + \sqrt{8n\ln\left(\frac{1}{\delta}\right)}\right)\sqrt{u\left(\kappa_2 + \lambda_1\right)}\sqrt{u\left(\kappa_2 + \lambda_1\right) + 8\lambda_1\kappa_3\kappa_5}}{4\lambda_1(n-1)} + \kappa_5\sqrt{\frac{\ln\left(\frac{1}{\delta}\right)}{2n}} \quad (23)$$

$$+ \left(1 + \sqrt{8n\ln\left(\frac{1}{\delta}\right)}\right)\left(\frac{u\sigma^{\mathrm{selec}}\sigma^{\mathrm{score}}\kappa_2\kappa_4^2\left(\kappa_2 + \lambda_1\right) + 4\lambda_1\sigma^{\mathrm{selec}}\kappa_2\kappa_3\kappa_5}{4\lambda_1(n-1)}\right),$$

where $\kappa_5 = \|((\Phi_* - \Phi_*^{\backslash j})\mathrm{W_E})\mathrm{W_D}\|_2$, $\sigma^{\mathrm{selec}} = \kappa_1\|(\Phi_*^{\max_k} + (\Phi_*^{\backslash j})^{\max_k})\mathrm{W_E}\mathrm{W_D}\|_2 + 2\kappa_1$, and $\sigma^{\mathrm{score}} = \kappa_1\|(\Phi_* + \Phi_*^{\backslash j})\mathrm{W_E}\mathrm{W_D}\|_2 + 2\kappa_1$.

*Further, the convergence rate of the above generalization error bounds is*

$$\mathcal{O}\left(\frac{\sqrt{\ln\left(\frac{1}{\delta}\right)}}{\sqrt{n}\min\{\sqrt{\lambda_1}, \lambda_1\}} + \sqrt{\frac{\ln\left(\frac{1}{\delta}\right)}{n}}\right).$$

*Proof.* By Theorems 1, we have that our feature selection algorithm is uniformly stable. Putting (14) into (6) of Theorem A, we have

$$L^{\mathrm{selec}}\left(\Phi_*^{\max_k}, S\right) - L_{emp}^{\mathrm{selec}}\left(\Phi_*^{\max_k}, S\right)$$

$$\leqslant 2\beta + (4n\beta + \kappa_5)\sqrt{\frac{\ln\left(\frac{1}{\delta}\right)}{2n}}$$

$$= \left(2 + 4n\sqrt{\frac{\ln\left(\frac{1}{\delta}\right)}{2n}}\right)\beta + \kappa_5\sqrt{\frac{\ln\left(\frac{1}{\delta}\right)}{2n}}$$

$$= \frac{\sigma^{\mathrm{selec}}\kappa_2\kappa_4\left(1 + \sqrt{2n\ln\left(\frac{1}{\delta}\right)}\right)\sqrt{u\left(\sigma^{\mathrm{selec}}\kappa_2 + \lambda_1\sigma^{\mathrm{score}}\right)}\sqrt{u\kappa_4^2\left(\sigma^{\mathrm{selec}}\kappa_2 + \lambda_1\sigma^{\mathrm{score}}\right) + 8\lambda_1\kappa_3\kappa_5}}{2\lambda_1(n-1)}$$

$$+ \left(1 + \sqrt{2n\ln\left(\frac{1}{\delta}\right)}\right)\left(\frac{u\sigma^{\mathrm{selec}}\kappa_2\kappa_4^2\left(\sigma^{\mathrm{selec}}\kappa_2 + \lambda_1\sigma^{\mathrm{score}}\right) + 4\lambda_1\sigma^{\mathrm{selec}}\kappa_2\kappa_3\kappa_5}{2\lambda_1(n-1)}\right)$$

$$+ \kappa_5\sqrt{\frac{\ln\left(\frac{1}{\delta}\right)}{2n}}.$$

---

[5]In the main paper, due to space limitations, we do not present the upper bound for $L^{\mathrm{selec}}(\Phi_*^{\max_k}, S) - L_{loo}^{\mathrm{selec}}(\Phi_*^{\max_k}, S)$.

By Lemma B, we have

$$L^{\text{selec}}\left(\Phi_*^{\max_k}, S\right) - L_{emp}^{\text{selec}}\left(\Phi_*^{\max_k}, S\right)$$

$$\leqslant \frac{\sigma^{\text{selec}}\sigma^{\text{score}}\kappa_2\kappa_4^2\left(1 + \sqrt{2n\ln\left(\frac{1}{\delta}\right)}\right)\sqrt{u\left(\kappa_2 + \lambda_1\right)}\sqrt{u\left(\kappa_2 + \lambda_1\right) + 8\lambda_1\kappa_3\kappa_5}}{2\lambda_1(n-1)} + \kappa_5\sqrt{\frac{\ln\left(\frac{1}{\delta}\right)}{2n}}$$

$$+ \left(1 + \sqrt{2n\ln\left(\frac{1}{\delta}\right)}\right)\left(\frac{u\sigma^{\text{selec}}\sigma^{\text{score}}\kappa_2\kappa_4^2\left(\kappa_2 + \lambda_1\right) + 4\lambda_1\sigma^{\text{selec}}\kappa_2\kappa_3\kappa_5}{2\lambda_1(n-1)}\right).$$

And plugging (14) into 7 of Theorem A, we have

$$L^{\text{selec}}\left(\Phi_*^{\max_k}, S\right) - L_{loo}^{\text{selec}}\left(\Phi_*^{\max_k}, S\right)$$

$$\leqslant \beta + (4n\beta + \kappa_5)\sqrt{\frac{\ln\left(\frac{1}{\delta}\right)}{2n}}$$

$$= \left(1 + 4n\sqrt{\frac{\ln\left(\frac{1}{\delta}\right)}{2n}}\right)\beta + \kappa_5\sqrt{\frac{\ln\left(\frac{1}{\delta}\right)}{2n}}$$

$$= \frac{\sigma^{\text{selec}}\kappa_2\kappa_4\left(1 + \sqrt{8n\ln\left(\frac{1}{\delta}\right)}\right)\sqrt{u\left(\sigma^{\text{selec}}\kappa_2 + \lambda_1\sigma^{\text{score}}\right)}\sqrt{u\kappa_4^2\left(\sigma^{\text{selec}}\kappa_2 + \lambda_1\sigma^{\text{score}}\right) + 8\lambda_1\kappa_3\kappa_5}}{4\lambda_1(n-1)}$$

$$+ \left(1 + \sqrt{8n\ln\left(\frac{1}{\delta}\right)}\right)\left(\frac{u\sigma^{\text{selec}}\kappa_2\kappa_4^2\left(\sigma^{\text{selec}}\kappa_2 + \lambda_1\sigma^{\text{score}}\right) + 4\lambda_1\sigma^{\text{selec}}\kappa_2\kappa_3\kappa_5}{4\lambda_1(n-1)}\right)$$

$$+ \kappa_5\sqrt{\frac{\ln\left(\frac{1}{\delta}\right)}{2n}}.$$

By Lemma B, we get

$$L^{\text{selec}}\left(\Phi_*^{\max_k}, S\right) - L_{loo}^{\text{selec}}\left(\Phi_*^{\max_k}, S\right)$$

$$\leqslant \frac{\sigma^{\text{selec}}\sigma^{\text{score}}\kappa_2\kappa_4^2\left(1 + \sqrt{8n\ln\left(\frac{1}{\delta}\right)}\right)\sqrt{u\left(\kappa_2 + \lambda_1\right)}\sqrt{u\left(\kappa_2 + \lambda_1\right) + 8\lambda_1\kappa_3\kappa_5}}{4\lambda_1(n-1)} + \kappa_5\sqrt{\frac{\ln\left(\frac{1}{\delta}\right)}{2n}} \qquad (24)$$

$$+ \left(1 + \sqrt{8n\ln\left(\frac{1}{\delta}\right)}\right)\left(\frac{u\sigma^{\text{selec}}\sigma^{\text{score}}\kappa_2\kappa_4^2\left(\kappa_2 + \lambda_1\right) + 4\lambda_1\sigma^{\text{selec}}\kappa_2\kappa_3\kappa_5}{4\lambda_1(n-1)}\right).$$

Thus, the proof is completed. $\qquad\square$

Let $((\Phi_*^j)^{\max_k}, \Phi_*^j)$ correspond to the optimal feature selector and feature scorer for the following error:

$$RL_{emp}^j(\Phi, S) \triangleq \frac{1}{n} \sum_{\mathrm{x}_i \in S^j} \ell^{\mathrm{selec}}(\Phi^{\max_k}, \mathrm{x}_i) + \lambda_1 \ell^{\mathrm{score}}(\Phi, \mathrm{x}_i), \tag{25}$$

where $j \in \{1, \ldots, n\}$.

Next, we prove Corollary A which bounds $\ell^{\mathrm{selec}}(\Phi_*^{\max_k}, \cdot) - \ell^{\mathrm{selec}}((\Phi_*^j)^{\max_k}, \cdot)$.

**Corollary A.** *Under Assumptions 1, 2, and 3, we have,* $\forall n \geqslant 2$,

$$\left\| \ell^{\mathrm{selec}}\left(\Phi_*^{\max_k}, \cdot\right) - \ell^{\mathrm{selec}}\left(\left(\Phi_*^j\right)^{\max_k}, \cdot\right) \right\|_\infty$$

$$\leqslant \frac{\sigma^{\mathrm{selec}}\sigma^{\mathrm{score}}\kappa_2\kappa_4^2\sqrt{u(\kappa_2+\lambda_1)}\sqrt{u(\kappa_2+\lambda_1)+8\lambda_1\kappa_3\kappa_5} + u\sigma^{\mathrm{selec}}\sigma^{\mathrm{score}}\kappa_2\kappa_4^2(\kappa_2+\lambda_1)}{2\lambda_1(n-1)} \tag{26}$$

$$+ \frac{2\sigma^{\mathrm{selec}}\kappa_2\kappa_3\kappa_5}{(n-1)},$$

*where* $\kappa_5 = \|((\Phi_* - \Phi_*^{\backslash j})\mathrm{W_E})\mathrm{W_D}\|_2$, $\sigma^{\mathrm{selec}} = \kappa_1\|(\Phi_*^{\max_k} + (\Phi_*^{\backslash j})^{\max_k})\mathrm{W_E}\mathrm{W_D}\|_2 + 2\kappa_1$, *and* $\sigma^{\mathrm{score}} = \kappa_1\|(\Phi_* + \Phi_*^{\backslash j})\mathrm{W_E}\mathrm{W_D}\|_2 + 2\kappa_1$.

*Proof.* Corollary A follows by noticing the following fact:

$$\left\| \ell^{\mathrm{selec}}\left(\Phi_*^{\max_k}, \cdot\right) - \ell^{\mathrm{selec}}\left(\left(\Phi_*^j\right)^{\max_k}, \cdot\right) \right\|_\infty$$

$$\leqslant \left\| \ell^{\mathrm{selec}}\left(\Phi_*^{\max_k}, \cdot\right) - \ell^{\mathrm{selec}}\left(\left(\Phi_*^{\backslash j}\right)^{\max_k}, \cdot\right) \right\|_\infty$$

$$+ \left\| \ell^{\mathrm{selec}}\left(\left(\Phi_*^{\backslash j}\right)^{\max_k}, \cdot\right) - \ell^{\mathrm{selec}}\left(\left(\Phi_*^j\right)^{\max_k}, \cdot\right) \right\|_\infty.$$

$\square$

Next, we prove Proposition 1 which reveals the relationship between the regularization and the perturbation of $L_{emp}^{\text{score}}(\Phi_*, S)$ and $L_{emp}^{\text{score}, \backslash j}(\Phi_*, S)$.

**Proposition 1.** *Let* $\Delta\Phi^{\max_k} \triangleq (\Phi_*^{\backslash j})^{\max_k} - \Phi_*^{\max_k}$, *and* $\Delta\Phi \triangleq \Phi_*^{\backslash j} - \Phi_*$. $\forall t \in [0, 1]$, *the following inequality holds:*

$$L_{emp}^{\text{score}}(\Phi_*, S) - L_{emp}^{\text{score}}(\Phi_* + t\Delta\Phi, S) + L_{emp}^{\text{score}, \backslash j}(\Phi_*, S) - L_{emp}^{\text{score}, \backslash j}(\Phi_* + t\Delta\Phi, S)$$

$$\leqslant \frac{t\kappa_1 \left( \left\| \left( 2\left(\Phi_*^{\backslash j}\right)^{\max_k} - t\Delta\Phi^{\max_k} \right) W_E W_D \right\|_2 + 2 \right) \left\| ((x\Delta\Phi^{\max_k}) W_E) W_D \right\|_2}{\lambda_1}.$$

*Proof.* Note that $\ell^{\text{selec}}(\Phi^{\max_k}, x)$ is convex with respect to $\Phi^{\max_k}$, and so is $L_{emp}^{\text{selec}}(\Phi^{\max_k}, S)$. By Definition C, $\forall t \in [0, 1]$, we have,

$$L_{emp}^{\text{selec}}(\Phi_*^{\max_k} + t\Delta\Phi^{\max_k}, S) - L_{emp}^{\text{selec}}(\Phi_*^{\max_k}, S)$$

$$+ L_{emp}^{\text{selec}}\left(\left(\Phi_*^{\backslash j}\right)^{\max_k} - t\Delta\Phi^{\max_k}, S\right) - L_{emp}^{\text{selec}}\left(\left(\Phi_*^{\backslash j}\right)^{\max_k}, S\right)$$

$$\leqslant t\left(L_{emp}^{\text{selec}}\left(\left(\Phi_*^{\backslash j}\right)^{\max_k}, S\right) - L_{emp}^{\text{selec}}(\Phi_*^{\max_k}, S)\right) \tag{27}$$

$$+ t\left(L_{emp}^{\text{selec}}(\Phi_*^{\max_k}, S) - L_{emp}^{\text{selec}}\left(\left(\Phi_*^{\backslash j}\right)^{\max_k}, S\right)\right)$$

$$\leqslant 0.$$

Furthermore, because $(\Phi_*^{\max_k}, \Phi_*)$ and $((\Phi_*^{\backslash j})^{\max_k}, \Phi_*^{\backslash j})$ are respectively the optimal feature selectors and feature scorers for (12) and (13), we have

$$RL_{emp}(\Phi_*, S) - RL_{emp}(\Phi_* + t\Delta\Phi, S) \leqslant 0, \tag{28}$$

and

$$RL_{emp}^{\backslash j}\left(\Phi_*^{\backslash j}, S\right) - RL_{emp}^{\backslash j}\left(\Phi_*^{\backslash j} - t\Delta\Phi, S\right) \leqslant 0. \tag{29}$$

Summing the left-hand sides of (28) and (29), we have

$$RL_{emp}(\Phi_*, S) - RL_{emp}(\Phi_* + t\Delta\Phi, S) + RL_{emp}^{\backslash j}\left(\Phi_*^{\backslash j}, S\right) - RL_{emp}^{\backslash j}\left(\Phi_*^{\backslash j} - t\Delta\Phi, S\right)$$

$$= \frac{1}{n} \sum_{i=1}^{n} \ell^{\text{selec}}(\Phi_*^{\max_k}, x_i) + \lambda_1 \ell^{\text{score}}(\Phi_*, x_i)$$

$$- \frac{1}{n} \sum_{i=1}^{n} \ell^{\text{selec}}(\Phi_*^{\max_k} + t\Delta\Phi^{\max_k}, x_i) - \lambda_1 \ell^{\text{score}}(\Phi_* + t\Delta\Phi, x_i) \tag{30}$$

$$\frac{1}{n} \sum_{i=1, i \neq j}^{n} \ell^{\text{selec}}((\Phi_*^{\backslash j})^{\max_k}, x_i) + \lambda_1 \ell^{\text{score}}\left(\Phi_*^{\backslash j}, x_i\right)$$

$$- \frac{1}{n} \sum_{i=1, i \neq j}^{n} \ell^{\text{selec}}\left(\left(\Phi_*^{\backslash j}\right)^{\max_k} - t\Delta\Phi^{\max_k}, x_i\right) - \lambda_1 \ell^{\text{score}}\left(\Phi_*^{\backslash j} - t\Delta\Phi, x_i\right).$$

Plugging (27) into (30), we have

$$\ell^{\mathrm{selec}}\left(\left(\Phi_*^{\backslash j}\right)^{\max_k}-t\Delta\Phi^{\max_k},\mathrm{x}_i\right)-\ell^{\mathrm{selec}}\left(\left(\Phi_*^{\backslash j}\right)^{\max_k},\mathrm{x}_i\right)$$

$$\leqslant \quad \sum_{i=1}^{n}\left(\lambda_1\ell^{\mathrm{score}}\left(\Phi_*+t\Delta\Phi,\mathrm{x}_i\right)-\lambda_1\ell^{\mathrm{score}}\left(\Phi_*,\mathrm{x}_i\right)\right) \tag{31}$$

$$+ \sum_{i=1,i\neq j}^{n}\left(\lambda_1\ell^{\mathrm{score}}\left(\Phi_*^{\backslash j}-t\Delta\Phi,\mathrm{x}_i\right)-\lambda_1\ell^{\mathrm{score}}\left(\Phi_*^{\backslash j},\mathrm{x}_i\right)\right).$$

And by Lemma A, we have

$$\sum_{i=1}^{n}\left(\ell^{\mathrm{score}}\left(\Phi_*,\mathrm{x}_i\right)-\ell^{\mathrm{score}}\left(\Phi_*+t\Delta\Phi,\mathrm{x}_i\right)\right)$$

$$+ \sum_{i=1,i\neq j}^{n}\left(\ell^{\mathrm{score}}\left(\Phi_*^{\backslash j},\mathrm{x}_i\right)-\ell^{\mathrm{score}}\left(\Phi_*^{\backslash j}-t\Delta\Phi,\mathrm{x}_i\right)\right)$$

$$\leqslant \quad t\left(\|\mathrm{x}\|_2\left\|\left(2\left(\Phi_*^{\backslash j}\right)^{\max_k}-t\Delta\Phi\right)\mathrm{W}_{\mathrm{E}}\mathrm{W}_{\mathrm{D}}\right\|_2+2\|\mathrm{x}\|_2\right)\frac{\|\left(\left(\mathrm{x}\Delta\Phi^{\max_k}\right)\mathrm{W}_{\mathrm{E}}\right)\mathrm{W}_{\mathrm{D}}\|_2}{\lambda_1}.$$

Finally, note that

$$L_{emp}^{\mathrm{score}}(\Phi_*,S)-L_{emp}^{\mathrm{score}}(\Phi_*+t\Delta\Phi,S)+L_{emp}^{\mathrm{score},\backslash j}(\Phi_*,S)-L_{emp}^{\mathrm{score},\backslash j}(\Phi_*+t\Delta\Phi,S)$$

$$= \quad \sum_{i=1}^{n}\left(\ell^{\mathrm{score}}\left(\Phi_*,\mathrm{x}_i\right)-\ell^{\mathrm{score}}\left(\Phi_*+t\Delta\Phi,\mathrm{x}_i\right)\right)$$

$$+ \sum_{i=1,i\neq j}^{n}\left(\ell^{\mathrm{score}}\left(\Phi_*^{\backslash j},\mathrm{x}_i\right)-\ell^{\mathrm{score}}\left(\Phi_*^{\backslash j}-t\Delta\Phi,\mathrm{x}_i\right)\right),$$

then we complete the proof. $\qquad\square$

## 10 Reconstruction and Classification Results versus $k$

By varying the number of selected features on ISOLET, we obtain the corresponding linear reconstruction errors and classification accuracy rates as the outputs from different algorithms. We plot the linear reconstruction errors in MSE and classification accuracy rates in Figure 8.

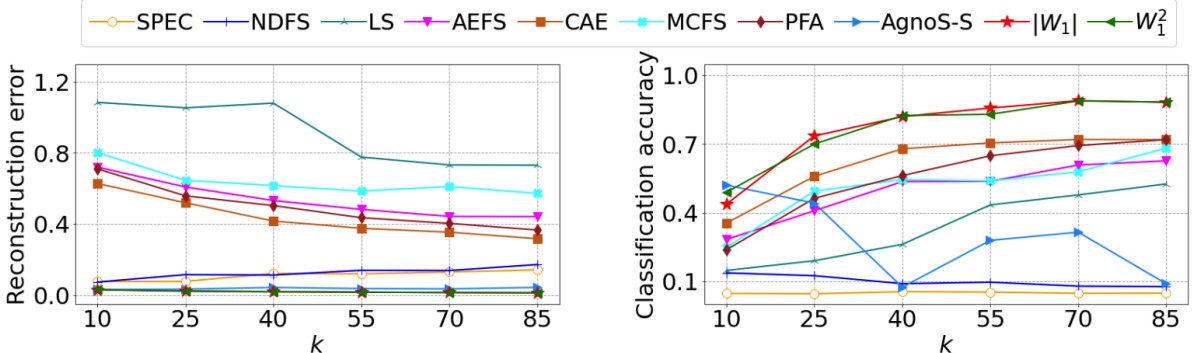

**Figure 8:** Reconstruction and classification results vs. the number of selected features on ISOLET.

## 11 Variational Accuracy with Reduction of Original Features

We take $\Phi^{\max_k} = |\mathrm{W_I}|^{\max_k}$, and we compare the variation of classification accuracy with the reduction of original features. The results are shown in Table 1 and Figure 9.

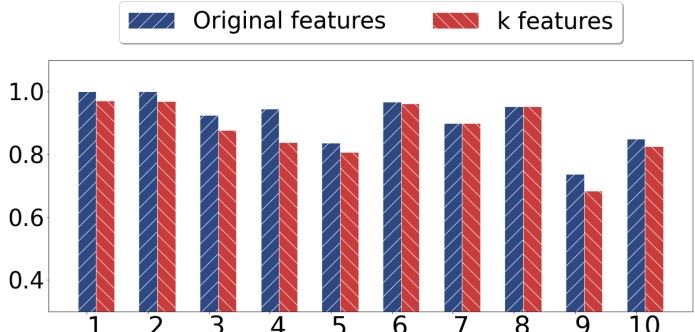

**Figure 9:** Comparison of classification results on original features and $k$ features over the 10 datasets. Here, $k$ can be obtained as $n \cdot (1$-reduction of #feature $/100)$, with $n$ being the #original features, and the reduction of #features can be found in Table 1.

**Table 1:** Reduction of classification accuracy (%) with the reduction of features (%) by our algorithm. Here, "↓ #Feature" denotes the eliminated proportion of features, and "↓ Accuracy" denotes the drop in classification accuracy. For example, on dataset USPS, with the selected features by our algorithm the number of features reduces by 80.5% (thus, the large majority of the original features are eliminated); the classification accuracy reduces only by 0.6%.

| Dataset No. | 1 | 2 | 3 | 4 | 5 | 6 | 7 | 8 | 9 | 10 | Average |
|---|---|---|---|---|---|---|---|---|---|---|---|
| ↓ #Features | 87.0 | 87.5 | 91.1 | 91.9 | 93.6 | 80.5 | 98.6 | 98.9 | 99.7 | 99.4 | 92.8±6.2 |
| ↓ Accuracy | 2.8 | 3.1 | 5.2 | 11.2 | 3.5 | 0.6 | 0.0 | 0.0 | 7.2 | 2.9 | 3.7±3.3 |

## 12   More Experiments of Algorithmic Stability

Adopting the same experiment design on COIL-20 in Section 6 of the main text, we vary $n$ from 100 to 900 with a step size of 200 to obtain different $S$. We delete a sample for each $S$ to get the corresponding $S^{\backslash i}$ and then calculate the left-hand side of (14) on the testing set for the trained models. From the plots in Figure 10 (a)-(b), it is seen that, since we compute the results by using the testing set instead of all the potential samples, we are subject to the interference from possible noise or outlier samples; however, with the increase of $n$, the curve of the uniform stability bound basically presents a downward tendency, which is consistent with our theoretical analysis.

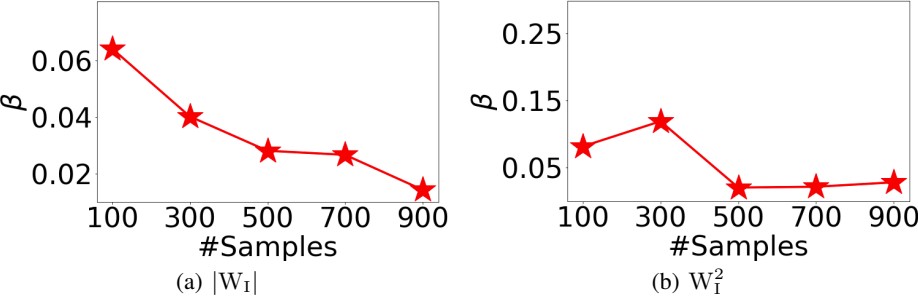

(a) $|W_I|$ (b) $W_I^2$

**Figure 10:** Algorithmic stability analysis on COIL-20 when $k = 50$.

## 13   More Results about Stability of Selected Features

We empirically analyze the stability of features selected by Eq. (2) of the main text. We randomly split the samples of COIL-20 into the training and testing sets, then use Eq. (2) to perform feature selection. We repeat this procedure 10 times with different random seeds and plot the selection results in Figure 11 (a)-(b). Note that the selected features essentially overlap for these 10 different splits and are stable.

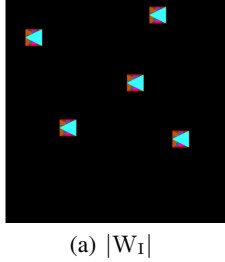    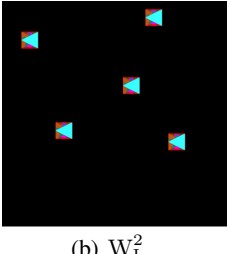

(a) $|W_I|$    (b) $W_I^2$

**Figure 11:** Stability analysis of 5 selected features. Note that 10 different splits yield essentially overlapping features.

We give the selected features from different splits together with original samples of MNIST-Fashion and COIL-20 in Figure 12 below.

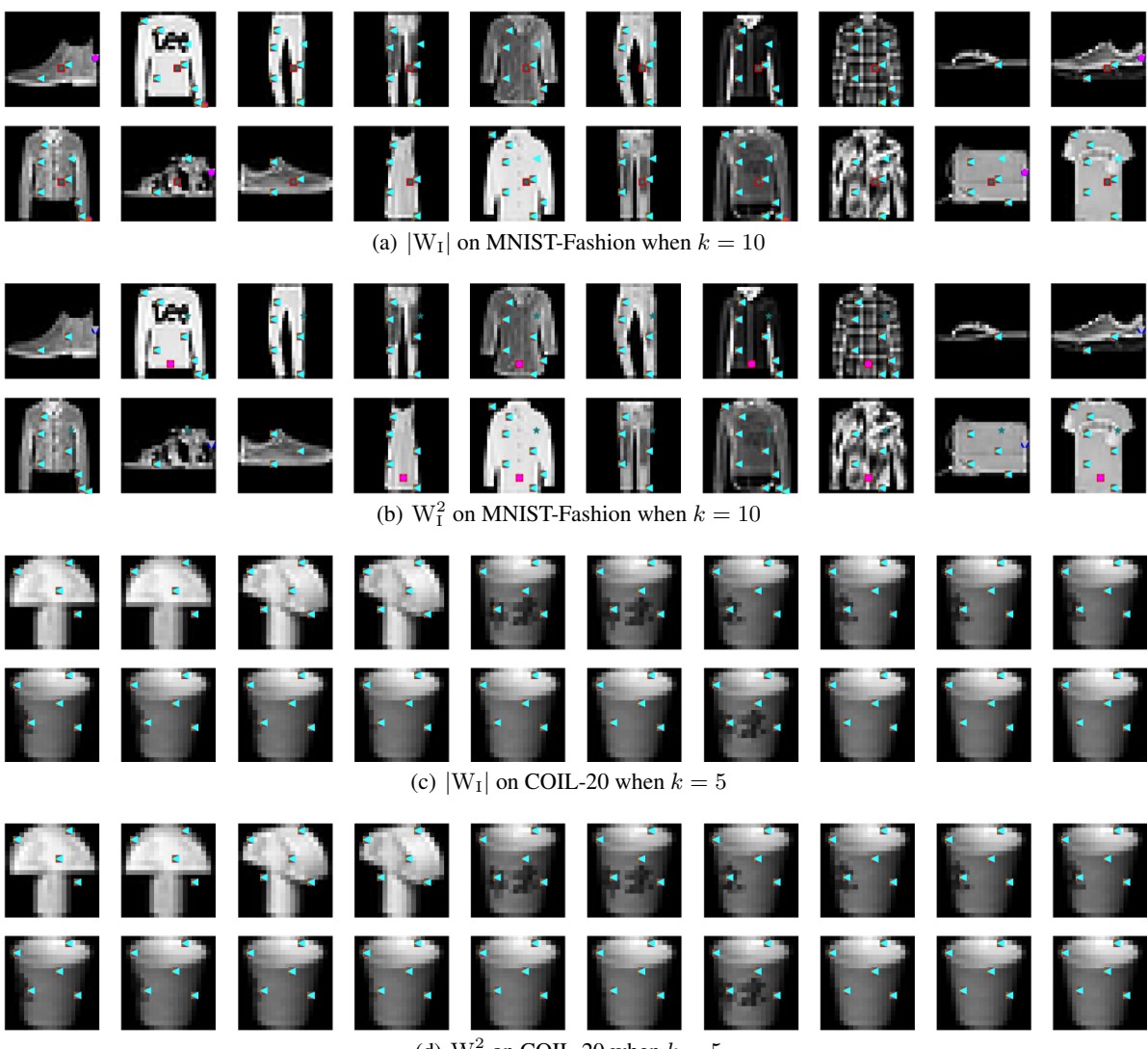

(a) $|W_I|$ on MNIST-Fashion when $k = 10$

(b) $W_I^2$ on MNIST-Fashion when $k = 10$

(c) $|W_I|$ on COIL-20 when $k = 5$

(d) $W_I^2$ on COIL-20 when $k = 5$

**Figure 12:** Selected key features on original samples.