# OpenReview forum: "Algorithmic stability and generalization of an unsupervised feature selection algorithm"
_NeurIPS.cc/2021/Conference — NeurIPS 2021 Poster_

### Official Review · Reviewer_LsED · 2021-07-13

**Rating:** 6
**Confidence:** 3

**Summary:**

The authors propose a novel methodology for unsupervised feature selection. Their procedure is based on simultaneously training a feature importance scorer neural network and a feature selector neural network; both are embedded inside an encoder in an autoencoder. The overall network is trained by optimizing a regularized estimator, where the regularization parameter induces some algorithmic stability. The authors provide theoretical guarantees of the estimator in the form of upper-bounds for the algorithmic stability and generalization. Further, they demonstrate the effectiveness of their approach on multiple real datasets and empirically understand various features of their algorithm (e.g. sensitivity to choices of parameters) via demonstrations on synthetic data.

**Limitations And Societal Impact:**

The authors could do a better job pointing out some limitations of their approach and furture work. No negative societal impact that I can foresee.

**Main Review:**

Originality: yes, to be the best of my knowledge, the methods seem novel!

Quality: In general, I found the paper interesting to read and the technical/methodological contributions to have some potential with good empirical performance. Below I state some questions/comments/concerns (some of which may be naive)

 --- The regularization term in equation (2) is motivated by the need for algorithmic stability. Looking at Theorem 1, it appears than when lambda is set to 0, the bound is vacuous and hence asymptotically \lambda must be chosen as 1/sublinear(n). This leads me to have the following remarks:

a) is the source of instability the non-continuous max_k operation? this to me seems like the major issue?

b) related to a, would then a potential alleviation be to consider simply the second term and impose an $\ell_1$ penalty on the elements of \Phi(W_I)? Do you think this would also makes sense as an estimator? What are the advantages of your proposed method of the potentially naive procedure I just outlined?

c) I appreciate the authors description that the combination of the two terms does something global and local but their framework does deviate from standard inverse problems where there is fidelity to data and a complexity controlling regularization.

d) I recognize that this is very hard, but is there any sense of the existence of lower bounds for the algorithmic stability?

e) understanding algorithmic stability as a function of lambda: the authors look at test error as a function of lambda. I am curious to see the same plot for algorithmic stability. I suppose due to the connection between generalization and algorithmic stability, we expect the same behavior

---- As denoted in Figure 1,  the method is perhaps useful for highlighting key attributes of the image in pixel domain. Sometimes the interesting attributes are concepts that are not well characterized purely by just pixels. What do you propose in this setting? Somehow there it makes sense to look at XW where W is reduces to a particular subspace. It would be good for the authors to comment on the utility of their framework in a "discovery" type aspect for imaging data.

---- in addition to k, \lambda_1, the bottleneck dimension d must be chosen. How sensitive is the method to this?

----- How exactly do the two neural networks (scorer and selector interact) other than combining together to map to the latent space?  I am sure I'm missing something simple here.

Clarity: the writing is reasonable although there is much room for improvement as I outline:
-- "To address these issues, we propose a novel unsupervised feature selection algorithm with a proven
35 algorithmic stability guarantee in this paper." --> sounds less awkward if "in this paper" appears after the comma
-- "It comprises a NN" -> it comprises of a NN
-- "evaluate the representation ability of the highly scored features locally." -> "locally evaluate the representation...."
-- "revealed by theoretical analysis and the stability of the selected features by our algorithm" -> sounds awkward. rephrase.
-- lines 54--60, remove the numbers as they make the structure awkward.
-- "be linked with the generalization" -> linked with generalization
-- "proves the stability" -> prove stability
-- "the role of the regularization" -> "the role of regularization"
-- Line 134: do not start a sentence with a symbol.
--- Line 139: good to specify dimension of $\Phi$ and introduce the quantity. It also appears that \Phi(W) can just be combined into one symbol?
-- "Our new algorithm is to use NN for the optimization of" -> awkward phrase
-- What is the difference between X(\Phi(W)) and X\Phi(W)? seems like both are used and this seems confusing.
-- "we obtain the feature selector and use it to make feature selection on new samples during testing" ->> seems ambiguous. please be more precise here.
-- "manifested in our experiments." -> "demonstrated in our experiments"
-- "There has been neither a clue about the role of the regulation in the uniform stability." -> grammatically incorrect with spelling error
--- line 155-159: could appear right after equation (2)?
--- line 178: missing a star below \Phi
--- Experimental section: it would be good to get a short and clear description of the architecture. I was unclear about this even after reading supplementary.
--- Line 240, the weights are sampled between two very strange numbers. Is there a reason for this?
--- "We set \lambda to be 1/2^7" -> this appears very small?
-- "the uniform stability" -> remove the
--- "meanwhile, the selected features by the proposed algorithm show comparable performance to the original features." -> use additionally instead of meanwhile.
--- "with respect to the training sample size, the regularization, the number of selected features, and the uniform stability" -> "with respect to sample size,  regularization levels, the number of selected features, and uniform stability"


Significance: Yes, I think the methodology could be interesting given its superior performance empirically over other methods and favorable theoretical properties.

**Time Spent Reviewing:**

8

---

> ### Author Response · Authors · 2021-08-10
> **Response to Reviewer LsED. Thanks for your constructive comments. Your intriguing questions prompted us to think further.**
>
> We sincerely appreciate your taking valuable time in reviewing our paper. For your questions, we have responded one by one below.
>
> ---
> ---
>
> ## Question 1:
>
> > + *Is the source of instability the non-continuous $max_k$ operation? this to me seems like the major issue?*
>
> ## Response 1:
>
> + We understood that you raised this question to doubt whether it would be an unstable method to use only the first term with the $\max_k$ operation, i.e., when $\lambda_1=0$. This is an intriguing question, but there appears to have no definite answer to it based on the mathematical results so far. While we also suspected the non-continuous $\max_k$ operation may have a connection with instability, we have no theoretical results or proof for this relationship. To fully address this question, we believe that more theoretical results need to be developed (to prove or disprove), which we feel would be out of the scope of this paper.
>
>
> + For the proposed method and as a part of theoretical results, we mathematically proved that our algorithm is algorithmically stable when $\lambda_1\neq 0$. The empirical experiments on various real-world data also validated these theoretical results. In our algorithm, we do not allow $\lambda_1=0$ so that the scorer (i.e., the second term) always works, which ensures the selected features by the selector have the largest importance.
>
> + We believe that the potential instability of the $\max_k$ operation would be an interesting topic to investigate in a different paper.
>
>
> $\quad $
>
> ---
> ---
>
> ## Question 2:
>
> > + *Related to the above quesion, would then a potential alleviation be to consider simply the second term and impose an $\ell$ penalty on the elements of $\Phi(W_I)$? Do you think this would also makes sense as an estimator? What are the advantages of your proposed method of the potentially naive procedure I just outlined?*
>
> ## Response 2:
>
> + This question is constructive. We think such a method for feature selection appears to be similar to the Lasso version in the supervised framework. From the theoretical point of view, $\ell$-regularization-based models are not algorithmically stable [Xu et al. 2011]. Our method is.
>
>      + *H. Xu, C. Caramanis, and S. Mannor. "Sparse algorithms are not stable: A no-free-lunch theorem." IEEE TPAMI, 34(1): 187-193, 2011.*
>
> $\quad $
>
> ---
> ---
>
> ## Question 3:
>
> > + *I appreciate the authors description that the combination of the two terms does something global and local but their framework does deviate from standard inverse problems where there is fidelity to data and a complexity controlling regularization.*
>
> ## Response 3:
>
> + We agree with your comment. Our design combined global and local considerations for feature selection, which is different from the traditional methods for inverse problems and is regarded as one of the novelties of this paper. The second term helps obtain $W_I$ to ensure the selected feature are most important and promote the first term to well approximate the input data $X$. By doing so, theoretically, the algorithmic stability of our proposed algorithm has a guarantee; experimentally, the effect of the combination was also validated.
>
> $\quad $
>
> ---
> ---
>
>
> ## Question 4:
>
> > + *I recognize that this is very hard, but is there any sense of the existence of lower bounds for the algorithmic stability?*
>
> ## Response 4:
>
> + For this thought-provoking question, currently we have no definite answer to it for the proposed algorithm. As far as we know, this is still an open question regarding the theoretical development of lower bounds for uniformly stable algorithms. [Feldman et al. 2019] asked whether it is possible to strengthen the existing high-probability upper generalization bounds and prove corresponding high probability lower bounds. [Bousquet et al. 2020] gave lower bounds based on some specific functions, but it is still elusive to fully answer the question of the optimality of their generalization bound for uniformly stable algorithms. We believe that this is a worthy topic that warrants future investigation.
>
>
>   + *V. Feldman and J. Vondrak. "High probability generalization bounds for uniformly stable algorithms with nearly optimal rate." In Conference on Learning Theory, pp. 1270-1279. PMLR, 2019.*
>
>   + *O. Bousquet, Y. Klochkov, and N. Zhivotovskiy. "Sharper bounds for uniformly stable algorithms." In Conference on Learning Theory, pp. 610-626. PMLR, 2020.*
>
>
> $\quad $
>
> ---
> ---
>
> ## Question 5:
>
> > + *Understanding algorithmic stability as a function of lambda: the authors look at test error as a function of lambda. I am curious to see the same plot for algorithmic stability. I suppose due to the connection between generalization and algorithmic stability, we expect the same behavior*
>
> >  *---- As denoted in Figure 1, the method is perhaps useful for highlighting key attributes of the image in pixel domain. Sometimes the interesting attributes are concepts that are not well characterized purely by just pixels. What do you propose in this setting? Somehow there it makes sense to look at XW where W is reduces to a particular subspace. It would be good for the authors to comment on the utility of their framework in a "discovery" type aspect for imaging data.*
>
> ## Response 5:
>
> + This question suggests the difference between feature selection and feature extraction (e.g., PCA). We believe our framework can be used in a "discovery" type aspect for imaging data as you pointed out. For example, we believe it would be useful in a multi-scale fashion to select other types of features/patterns for discovery in a similar way to the pyramid network (e.g., [Zhao et al. 2021]), so that the notion (i.e., the combination of a selector and a scorer) proposed in our paper might be applied to automatically select a super-pixel hierarchy $\mathcal{S}$ instead of manually defining $\mathcal{S}$.
>
>      + *G. Zhao, W. Ge, and Y. Yu. "GraphFPN: Graph Feature Pyramid Network for Object Detection." arXiv preprint arXiv:2108.00580 (2021).*
>
>
> $\quad $
>
> ---
> ---
>
> ## Question 6:
>
> > + *In addition to k, $\lambda_1$, the bottleneck dimension $d$ must be chosen. How sensitive is the method to this?*
>
> ## Response 6:
>
>
> + When choosing $\lambda_1$ too large or too small, the algorithm could not have a good performance (more discussions are provided on Lines 282-292, Page 8); for $k$, when it increases, the test error will decrease. For datasets 1-6, we followed the setting in Ref. [1], that is, $k=50$; for datasets 7-10, since they have more features than datasets 1-6, we empirically set $k=64$. On MNIST-Fashion, we tested the effect of different $k$ in Figure 2 (c); for $d$, during our empirical experiments, we found that the performance for the selected feature is quite insensitive as long as it is not too small. Several baseline methods, such as CAE and AgnoS-S, involve $d$ in the latent space (i.e., the dimension of the bottleneck). Thus, for a fair comparison, in the experiments of our paper, we set $d$ to be the same as the number of selected features $k$. In practical applications, in general, we can tune these parameters on validation sets.
>
> $\quad $
>
> ---
> ---
>
> ## Question 7:
>
> > + *How exactly do the two neural networks (scorer and selector interact) other than combining together to map to the latent space? I am sure I'm missing something simple here.*
>
> ## Response 7:
>
> + The feature scorer and selector will indeed interact and be estimated iteratively through our global- and sub-neural networks. Putting in details, Eq. (2) in our paper can be regarded as an extension of a common NN, with the main difference from the first term called 'selector' (the second term is 'scorer'). For the training based on Eq. (2), in each iteration of backpropagation,
>
>    + The selector will require the gradients of the features having the top-$k$ weights in magnitude for this iteration while having no effect on the gradients of other features. Thus, in each iteration, the gradients of the selector need to adopt a ranking operation to obtain the $k$-largest weights from the scorer. And the selector will update the weights of the corresponding $k$ selected features to ensure these features to well reconstruct the original input data $X$.
>
>    + The scorer updates the weights based on the backpropagation which includes the contribution from the selectors and, at the same time, re-scores the features for the selector to perform ranking and selecting the top-$k$ features.
>
>
> + After training, only the trained selector is used to select features and do reconstruction during testing time.
>
>
> $\quad $
>
> ---
> ---
>
>
> ## Question 8:
>
> > + *It would be good to get a short and clear description of the architecture. I was unclear about this even after reading supplementary.*
>
> ## Response 8:
>
>
> A more detailed description can be found in **Response 7**. Briefly speaking, during training, the sub-NN of the architecture, i.e., the selector, will select the top-$k$ features (from the scorer weights) by ensuring these selected features to well reconstruct the original input $X$ with the main NN. On the other hand, the scorer, as a constraint, will ensure the selected features for the selector to have the largest importances globally. We will add more explanation about the architecture of our algorithm in the future version.
>
>
> $\quad $
>
> ---
> ---
>
> ## Question 9:
>
> > + *Line 240, the weights are sampled between two very strange numbers. Is there a reason for this?*
>
> ## Response 9:
>
> Please see **Response 3** to **Question 3** of Reviewer 2.
>
> $\quad $
>
> ---
> ---
>
> ## Question 10:
>
> > + *"We set \lambda_1 to be 1/2^7" -> this appears very small?*
>
> ## Response 10:
>
> Please see **Response 4** to **Question 4** of Reviewer 2.
>
>
> In addition, we will fix all the editorial issues and typos you helped point out in the future version.

---

### Official Review · Reviewer_Fihd · 2021-07-14

**Rating:** 7
**Confidence:** 4

**Summary:**

The authors address the problem of unsupervised feature selection. They combine an autoencoder reconstruction loss with an additional term which only uses the k most informative features to reconstruct the input data. The k most informative features are identified by sorting the absolute values of a diagonal matrix whose weights are multiplied by the input features. The authors prove that the method is algorithmically stable and find its generalization error bound. The method demonstrates superior capabilities compared with several feature selection baselines.

**Limitations And Societal Impact:**

The proposed method could help reduce the computational burden of training ML algorithms. The main challenge in the proposed method is to tune the additional hyperparameters, namely k and \lambda_1.

**Main Review:**

The problem of unsupervised feature selection is important and has several applications in ML. The method proposed by the authors is simple but seems to work very well on a diverse set of datasets. The theoretical analysis provided by the authors is novel and sheds light on the stability and generalization capabilities of the method. The paper is well written and the presentation is clear. Overall I recommend accepting the paper, however, I have several comments which could improve the clarity of the results.
-P1L10 show our-> show that our
-P1L32 none -> no
-P2L44 an extra space appears before the word “It”
-Notation: You are using X’ from transpose but then you use x’ again to denote something else.
P3L131 the sentence is poorly written, do you mean: The goal of feature selection is…?
P7 Design of experiments: the reasoning for initializing the selection layer from the very narrow uniform distribution is not clear. Why not just initialize all of them as 1?
The selection of \lambda_1 is also not clear, did this value lead to optimal results on one dataset or on the average/median of all of them?
What is the batch size used?
How are the parameters of Extremely randomized trees tuned?
The relation between the method and Agnos-S (and perhaps other methods) should be clearly explained in a related work section.
 Was the linear regression (used for evaluating the MSE) also trained on the train and evaluated on the test?


**Time Spent Reviewing:**

10-20

---

> ### Author Response · Authors · 2021-08-10
> **Response to Reviewer Fihd. Thanks for your constructive comments. We will modify following your insightful suggestions.**
>
> We sincerely appreciate your taking valuable time in reviewing our paper. For your questions, we have responded one by one below.
>
> ---
> ---
>
> ## Question 1:
>
> > + *You are using X’ from transpose but then you use x’ again to denote something else.*
>
> ## Response 1:
>
> + Please see **Response 6** to **Question 6** of Reviewer 1.
>
>
> $\quad $
>
> ---
> ---
>
> ## Question 2:
>
> > + *P3L131 the sentence is poorly written, do you mean: The goal of feature selection is…?*
>
> ## Response 2:
>
> + Yes. We will revise and improve this sentence.
>
> $\quad $
>
> ---
> ---
>
> ## Question 3:
>
> > + *P7 Design of experiments: the reasoning for initializing the selection layer from the very narrow uniform distribution is not clear. Why not just initialize all of them as 1?*
>
> ## Response 3:
>
> + The reasons are as follows: We adopted this way of initialization to facilitate the selection at the sub-NN and the implementation of the whole model. At the beginning of running our feature selection algorithm, we would like to have all the weights in the selection layer equal because, before incorporating information from the data, all features are regarded as playing equal roles. However, if we initialize all the weights to be 1, then in the first step the sub-NN could not select the top-$k$ weights; to break the tie, one option is to have the sub-NN randomly select $k$ features in the first step. The adopted initialization is a convenient way to have (almost) equal weights and at the same time avoid random selection in the first step. It may be regarded as a “small trick” for initialization here. Briefly, it was designed to add small perturbations to 1 to help break the potential tie at the beginning.
>
> $\quad $
>
> ---
> ---
>
> ## Question 4:
>
> > + *The selection of $\lambda_1$ is also not clear, did this value lead to optimal results on one dataset or on the average/median of all of them?*
>
> ## Response 4:
>
> + $\lambda_1$ is a regularization parameter, which is tuned on the validation set. In this paper, we fine-tuned it on the validation set of MNIST-Fashion, and then we used the tuned value for other datasets in the spirit of transfer learning. As shown by the experimental results, superior performance can already be achieved. If tuning $\lambda_1$ individually on the validation sets of different datasets, it is expected to achieve better performance; yet, it would be more time-consuming.
>
> $\quad $
>
> ---
> ---
>
> ## Question 5:
>
> > + *What is the batch size used?*
>
> ## Response 5:
>
> + For batch sizes, we individually tuned it on the validation sets of different datasets for the case $|W_I|$, then we also used them for the case of $W_{I}^2$. For example, the batch size value was 64 for ISOLET, 16 for arcene, and so on.
>
> $\quad $
>
> ---
> ---
>
> ## Question 6:
>
> > + *How are the parameters of Extremely randomized trees tuned?*
>
> ## Response 6:
>
> + For the Extremely randomized trees, we used the function `ExtraTreesClassifier` in the library `sklearn.ensemble`. For a fair comparison, we followed the same experimental settings of Ref. [1]: The number of trees in the forest was set to 50, and other parameters were set to default values.
>
> $\quad $
>
> ---
> ---
>
> ## Question 7:
>
> > + *The relation between the method and Agnos-S (and perhaps other methods) should be clearly explained in a related work section.*
>
> ## Response 7:
>
> + We will provide a detailed explanation about the relation between Agnos-S and our method in the future version.
>
> $\quad $
>
> ---
> ---
>
> ## Question 8:
>
> > + *Was the linear regression (used for evaluating the MSE) also trained on the train and evaluated on the test?*
>
> ## Response 8:
>
> + Yes. The linear regression was trained on the training data and then evaluated on the test data. The split ratio of the training data and test data is described in Lines 232-237, Pages 6-7.
>
> Moreover, we will fix all the editorial issues and typos you helped point out.

---

> > ### Comment · Reviewer_Fihd · 2021-08-11
> > **Response to authors**
> >
> > I thank the authors for addressing all my concerns. This information must be incorporated in the revised version of the manuscript, including details about the batch size and parameters of extremely randomized trees. Furthemore, following Q1 of reviewer #1, the authors should clearly explain in the paper that some of the results of table 2 were borrowed from [1]. I keep my score at 7 and believe that the proposed method contributes to the ML community.

---

> > > ### Author Response · Authors · 2021-08-12
> > > **Response to Reviewer Fihd. Thanks.**
> > >
> > > Thanks for reading our responses.
> > >
> > > We sincerely appreciate your valuable comments and suggestions, and we shall follow and incorporate them in revising the paper.

---

### Official Review · Reviewer_4y4c · 2021-07-16

**Rating:** 5
**Confidence:** 4

**Summary:**

This paper proposes an unsupervised feature selection method. The paper provides theoretical analysis of selection performance when the underlying dataset is perturbed. It provides experimental results on multiple datasets, comparing the performance of the proposed approach with that of the methods in the literature.

**Limitations And Societal Impact:**

Yes.

**Main Review:**

-Originality: The paper seems novel.
-Quality: Quality could be significantly improved. (see below)
-Clarity: Clarity is significantly below the acceptable level. (see below)
-Significance: Low (see below)

I think the paper has two significant flaws:

1)
- Experiments appear highly biased against deep feature selection methods: reconstruction and classification experiments use linear approaches whereas those methods, such as AEFS, operate under the assumption that deep feature selection is allowed. Indeed, for instance on the ISOLET dataset, Ref. 14 reports a classification accuracy of 89.2% (Table 2). The corresponding linear performance reported in this manuscript (Table 3) is only 57.6%. Similarly, while f and g can be nonlinear in Eq. 2, they are *tailored* for the linear performance metrics by virtue of restricting them to the linear case.
- Related to this point, how do the authors explain the huge performance jump for AEFS between tasks 1-5 and tasks 6-10?
- Related to this point, it is concerning that the proposed method performs much better in tasks 1-5 compared to tasks 6-10 in terms of the recon. error. (Table 2) The manuscript mentions that comparison results for tasks 1-5 were imported from the CAE paper.

2) Theoretical development is not careful enough, making it very hard to follow the paper.

a. Notation is inconsistent and cumbersome. I point out a few examples here:
- line 116: The definition of S^i is not clear because the inputs are S, i, and the new value x_i’.
- line 112: The definition of \Delta^t requires 3 arguments; f, x, x’ because the right hand side evaluates f at x+t\Delta x.
- line 172: L^{\text selec} is not defined.
- line 119: \ell is defined on \cal{X}x\cal{X}. But, A_S is a function.
- x’ denotes a (different) sample whereas X’ denotes matrix transposition. Would the authors consider a different notation for either one?

b. I have similar concerns about the theoretical results. I point out a few examples here:
- (Lemma C) \ell^{score} and \ell_{selec} have two arguments. Def. D considers functions of a single argument.
- (Lemma C) Def. D requires the inequality to hold for any arguments in its domain, but Lemma C specifies these arguments.
- (Lemma D) How Prop. 1 is used in Lemma D is not clear at all.
- (Lemma E) In the proof, please add more details showing the second equality involving limits.

c. Small lambda_1 corresponds to little regularization, so it makes sense that the bound in Eq. 4 is not meaningful. However, when lambda_1 is large, the first term on the right hand side of Eq. 4 will produce a lambda_1-independent term. Is this term small enough to produce a meaningful bound?


**UPDATE:** After a long discussion with the authors, I decided to increase my score from 4 to 5.

**Time Spent Reviewing:**

~8 hours

---

> ### Author Response · Authors · 2021-08-10
> **Response to Reviewer 4y4c. Thanks for your detailed comments. We believe that our algorithm performs well and the proofs are reliable.**
>
> We sincerely appreciate your valuable time reviewing our paper. For your questions, we have responded one by one below.
>
> ---
> ---
>
> ## Question 1:
>
> > + *Experiments appear highly biased against deep feature selection methods: reconstruction and classification experiments use linear approaches whereas those methods, such as AEFS, operate under the assumption that deep feature selection is allowed. Indeed, for instance, on the ISOLET dataset, Ref. 14 reports a classification accuracy of 89.2% (Table 2). The corresponding linear performance reported in this manuscript (Table 3) is only 57.6%. Similarly, while f and g can be nonlinear in Eq. 2, they are tailored for the linear performance metrics by virtue of restricting them to the linear case.*
>
> > + *Related to this point, how do the authors explain the huge performance jump for AEFS between tasks 1-5 and tasks 6-10?*
>
> > + *Related to this point, it is concerning that the proposed method performs much better in tasks 1-5 compared to tasks 6-10 in terms of the recon. error. (Table 2) The manuscript mentions that comparison results for tasks 1-5 were imported from the CAE paper.*
>
> ## Response 1:
>
> + For the results of baseline models on datasets 1-5, we directly imported them from Ref. [1] which is a recent paper in ICML 2019. For a fair comparison, we directly used their results for the baseline models in comparison in our paper; we also followed the same experimental settings of Ref. [1], such as the data partitioning ratio and downstream learning models.
>
> + In our paper, the results in Table 2 are for reconstruction based on the subsets of selected features. By using a simple linear regression model, our algorithm already gives comparable or smaller reconstruction performance with the selected subsets of features than using all features on majority datasets. These results clearly indicate that our algorithm has a relatively stable ability to select a subset of representative features, as pointed out on Line 258. Note that our goal here is not to show the best possible reconstruction.
>
> + The results in Table 3 show that our algorithm exhibits almost consistently superior performance in the downstream classification task on diverse datasets. As pointed out above, we followed Ref. [1] and used extremely randomized trees as a downstream classifier. In general, the performance of the models on different datasets would be different. We have checked the algorithms (both baseline models and the proposed) in comparison. For example, the results of baseline models imported from Ref. [1] are:
>
>    + on dataset 3, LS (28.0%), UDFS (28.7%), MCFS (29.5%), and our algorithm (87.7%),
>
>    + on dataset 5, LS (51.7%), UDFS (54.7%), MCFS (51.3%), and our algorithm (80.8%);
>
>    while the results we computed are:
>
>    + on datasets 7, LS (50.0%), UDFS (70.0%), MCFS (40.0%), and our algorithm (90.0%),
>
>    + on dataset 10, LS (62.5%), UDFS (Nan), MCFS (62.5%), and our algorithm (82.5%).
>
>    We think the variation of the algorithms’ performance on different datasets should be a normal phenomenon; also, the trend appears to be similar for most compared algorithms. For example, from dataset 3 to dataset 7, LS has increased accuracy, and we can observe the increases for other algorithms.
>
> + The classification result of AEFS on ISOLET in Table 3 is the same as Ref. [1], because the results for datasets 1-5 were imported from Ref. [1]. You pointed out that the result of AEFS on ISOLET in Ref. [14] is 89.2% which is higher. One possible reason is that Ref. [14] used a different way/setting to perform experiments from Ref. [1] and our paper. For example, in Ref. [14] a different downstream classifier is used: Ref. [14] used the nearest neighbor classifier as a downstream classifier, in contrast to extremely randomized trees in Ref. [1] and our paper. Also, different numbers of selected features were used for the downstream classifier: Ref. [14] described that “we set the number of selected features as [50, 100, 150, 200, 250, 300] and report the best results from the optimal parameters for all the methods.” In contrast, the results on dataset ISOLET in Table 3 were obtained only with the number of selected features being 50, following Ref. [1].
>
> + As we stated in the paper (see Line166 on Page 4), the concise linear version of $f$ and $g$ in our proposed method already showed superior performance; yet, it is worth noting that our model is naturally extensible to highly nonlinear functions. As elaborating the extension to more, nonlinear cases would be beyond the scope of the current paper, we decide to leave the extension to highly nonlinear functions for future work.
>
> $\quad $
>
> ---
> ---
>
> ## Question 2:
>
> > + *line 116: The definition of $S^i$ is not clear because the inputs are $S$, $i$, and the new value $x_i’$.*
>
> ## Response 2:
>
> + Here we followed a common usage in learning theory, e.g., [Bousquet et al. 2002], and it denotes replacing the $i$-th element in the original $S$.
>
>   + *O. Bousquet and A. Elisseeff. "Stability and generalization." The Journal of Machine Learning Research, 2 (2002): 499-526.*
>
> $\quad $
>
> ---
> ---
>
> ## Question 3:
>
> > + *line 112: The definition of $\Delta^t$ requires 3 arguments; $f$, $x$, $x’$ because the right hand side evaluates $f$ at $x+t\Delta x$.*
>
> ## Response 3:
>
> + Here $x’$ is a function of $x$ and $\Delta x$.
>
> $\quad $
>
> ---
> ---
>
> ## Question 4:
>
> > + *line 172: $L^{\text selec}$ is not defined.*
>
> ## Response 4:
>
> + The definition of generalization error is given on Line 119, Page 3. The definition of $L^{\text selec}$ is similar to $L^{\text selec}_{emp}$, which is based on the definition of $\ell^{\text selec}$ on Line 142, Page 4. In the future version, we will add an explanation to make it clearer.
>
> $\quad $
>
> ---
> ---
>
> ## Question 5:
>
> > + *line 119: \ell is defined on \cal{X}x\cal{X}. But, A_S is a function.*
>
> ## Response 5:
>
> + Our paper focused on unsupervised learning, where the domain of $A_S$ belongs to $\cal{X}$. We adopted a common usage in learning theory [Bousquet et al. 2002].
>
> $\quad $
>
> ---
> ---
>
> ## Question 6:
>
> > + *$x’$ denotes a (different) sample whereas $X’$ denotes matrix transposition. Would the authors consider a different notation for either one?*
>
> ## Response 6:
>
> + We will modify $X’$ into $X^T$ in the future version.
>
> $\quad $
>
> ---
> ---
>
> ## Question 7:
>
> > + *(Lemma C) $\ell^{score}$ and $\ell_{selec}$ have two arguments. Def. D considers functions of a single argument.*
>
> ## Response 7:
>
> + Lemma C already stated that it is about the first variables of $\ell^{score}$ and $\ell_{selec}$. In order to make it easier for readers to understand, we will add an explanation in the future version.
>
> $\quad $
>
> ---
> ---
>
> ## Question 8:
>
> > + *(Lemma C) Def. D requires the inequality to hold for any arguments in its domain, but Lemma C specifies these arguments.*
>
> ## Response 8:
>
> + In Lemma C, the first variables (i.e., the first variables together with the coefficients as a whole) of $\ell^{score}$ and $\ell_{selec}$, can be regarded as squared functions; so it can be proved that Definition D is satisfied by any arguments in its domain. A similar technique was also used in Ref. [23].
>
> $\quad $
>
> ---
> ---
>
> ## Question 9:
>
> > + *(Lemma D) How Prop. 1 is used in Lemma D is not clear at all.*
>
> ## Response 9:
>
> + It is a typo and it should be Property A, not Property 1. The proof of Lemma D needs a property of the Bregman Divergence, i.e., Property A. We will revise it.
>
> $\quad $
>
> ---
> ---
>
> ## Question 10:
>
> > + *(Lemma E) In the proof, please add more details showing the second equality involving limits.*
>
> ## Response 10:
>
> + Thanks for your suggestion. We will add more details to make it easier to understand.
>
> $\quad $
>
> ---
> ---
>
> ## Question 11:
>
> > + *Small lambda_1 corresponds to little regularization, so it makes sense that the bound in Eq. 4 is not meaningful. However, when lambda_1 is large, the first term on the right hand side of Eq. 4 will produce a lambda_1-independent term. Is this term small enough to produce a meaningful bound?*
>
> ## Response 11:
>
> + We disagree. The bound in Eq. (4) is mainly about the approximate behavior of $n$. As a regularization parameter, $\lambda_1$ has to be chosen and used as a constant with respect to $n$. Generally speaking, $\lambda_1$ is tuned and chosen on validation data in the practical computation.
>
> + If $\lambda_1$ is too large, then the first term of our model, i.e., the selector, will not work well.
>
>
> We sincerely hope that our response and explanation above have cleared up the ambiguity. In addition, we will fix all the typos you helped point out.

---

> > ### Comment · Reviewer_4y4c · 2021-08-20
> > **important experimental flaws + questionable theory**
> >
> > I would like to reiterate that I find the experimental setup very problematic: the deep networks are instructed to choose the features that are most capable of accurate reconstruction using potentially nonlinear transformations, but the tabulated accuracy test is based on linear methods. Results being imported from previous publications for some of the datasets does not address my concern at all.
> >
> > I would also like to reiterate that it’s not clear if Eq. 4 will ever provide a meaningful bound. When the problem is over-regularized (large lambda_1), at the expense of overall performance, one should be able to infer a meaningful bound on stability. I’m not sure if this is currently the case, and it is not easy to find a regime if/when this bound is insightful/meaningful. More generally, I find the math presented in a way that makes it very hard to follow and severely limits insights.
> >
> > I decided to keep my score at its current level.

---

> > > ### Author Response · Authors · 2021-08-21
> > > **Response to Reviewer 4y4c. We believe our experiment settings are clear and standard, and our theoretical results are reliable.**
> > >
> > > We sincerely appreciate your valuable time reviewing our responses. For your questions, we have responded as follows, hoping to clarify your uncertainty and misunderstanding.
> > >
> > > ---
> > > ---
> > >
> > > ## Question 1:
> > >
> > > > + *I would like to reiterate that I find the experimental setup very problematic: the deep networks are instructed to choose the features that are most capable of accurate reconstruction using potentially nonlinear transformations, but the tabulated accuracy test is based on linear methods. Results being imported from previous publications for some of the datasets does not address my concern at all.*
> > >
> > >
> > >
> > >
> > > ## Response 1:
> > >
> > >  + It is a misunderstanding. The experimental results and the implementation codes of AEFS, PFA, and CAE on datasets 1-5 were adopted from Ref. [1]; for a fair comparison, we also used the similar experimental setting on datasets 6-10 (except with a different $k$) and the same two metrics as Ref. [1], that is, a linear regression model for reconstruction and extremely randomized trees for classification, to evaluate the performance of feature selection. We believe that the base architectures and metrics are reasonable or standard, since the architectures of algorithms in comparison, for instance, AEFS, used the same standard **shallow** autoencoder with a single hidden layer as Ref. [1].
> > >
> > > + Then, for downstream models, either linear regression or extremely randomized trees, after feature selection, firstly we split the data with reduced dimensions into training and test data; then we trained these models on the training data and evaluated on the test data. Such a way of evaluation was commonly used for unsupervised feature selection in the literature, for example, in Ref. [1]. Also, we restate that the results in Table 2 mainly demonstrated that our algorithm has a stable ability to select a subset of representative features on majority datasets, and the results in Table 3 showed that our algorithm exhibits almost consistently superior performance in the downstream classification task with extremely randomized trees on diverse datasets, as pointed out on Lines 257-258 of Page 7.
> > >
> > > + It is noted that the unsupervised feature selection models which depend on NNs, for example, AEFS, AgnoS-S, and our proposed algorithm, used a standard **shallow** autoencoder with a single hidden layer (i.e., the bottleneck); that is, we did not adopt deep architectures. We directly adopted the implementation codes of AEFS and CAE from Ref. [1], and used similar architectures for implementing our proposed algorithm.
> > >
> > >
> > > Moreover, we would like to point out that we have provided the main codes of the algorithms in comparison in the supplementary material. The experimental setup was similarly adopted in these algorithms.
> > >
> > >
> > >
> > > $\quad $
> > >
> > > ---
> > > ---
> > >
> > > ## Question 2:
> > >
> > > > + *I would also like to reiterate that it’s not clear if Eq. 4 will ever provide a meaningful bound. When the problem is over-regularized (large lambda_1), at the expense of overall performance, one should be able to infer a meaningful bound on stability. I’m not sure if this is currently the case, and it is not easy to find a regime if/when this bound is insightful/meaningful. More generally, I find the math presented in a way that makes it very hard to follow and severely limits insights.*
> > >
> > > ## Response 2:
> > >
> > >
> > > + It is a misunderstanding. We restate that the bound in Eq. (4) is to reveal the relationship between the deviation of the loss due to data perturbations and the varying $n$, i.e., the number of samples. This is a starting point for showing algorithmic stability and also an important topic in learning theory; see Refs. [8] and [Xu et al. 2011]. Note that  $\lambda_1$ in Eq. (4) is a regularization parameter.
> > >
> > >    *H. Xu, C. Caramanis, and S. Mannor. "Sparse algorithms are not stable: A no-free-lunch theorem." IEEE TPAMI, 34(1): 187- 193, 2011.*
> > >
> > > + As a regularization parameter, when $\lambda_1$ is chosen, from our theoretic bound and the discussion in Ref. [8] for the algorithmic stability on a regularization parameter $\lambda_1$, it is guaranteed that, as long as $\lambda_1$ is larger than a small quantity, for instance, $\lambda_1>1/n$ in Eq. (4), then we can obtain stability certified learning results. It is noted that $\lambda_1$ is not taken as a variable as $n$ for the bound.
> > >
> > > + For a regularization parameter like $\lambda_1$, as a common practice, it is not taken as a variable like $n$ in machine learning algorithms, and it is needed to be chosen based on validation data (Previously, we pointed it out in **Response 11** to your **Question 11**) or a priori information such as a user's experience. Once the tuning has been done on the validation data, it generally should not be changed on the test data.
> > >
> > > + Empirically, for $\lambda_1$, in our paper we carried out several experiments to further study its effect; see Lines 282-292 on Page 8. By experiments, we clearly pointed out that if $\lambda_1$ is chosen too large, it will over-emphasize the scorer, thus making the feature selector underfit training samples and incurring larger test errors. For example, as you pointed out that if we considered a large $\lambda_1$, then the validation error should be large. Therefore, such a $\lambda_1$ would never be chosen. In all, your statement of "*...When the problem is overregularized (large $\lambda_1$) ... one should be able to infer a meaningful bound on stability*" does not hold.
> > >
> > > In addition, you mentioned that "*the math presented in a way that makes it very hard to follow*". Actually, we elaborated the mathematical derivations and proofs in a step-by-step fashion in the supplementary material for readers to easily follow; to obtain necessary theoretical guarantees for our novel algorithm, a certain number of mathematical formulas and derivations are hard to avoid. If you would like to specify those relevant parts that are hard to follow, then it would be our great pleasure to explain them in more detail so that they would be clear to you. We would appreciate a lot your specifying those places that you thought were unclear so that we can clarify them.

---

> > > > ### Comment · Reviewer_4y4c · 2021-08-24
> > > > **multiple runs**
> > > >
> > > > Following the authors’ response and a discussion with the other reviewers and AC, my concern regarding lambda_1 is essentially resolved - thanks. However, I still have doubts regarding the experimental setup: the reported results for the proposed algorithm corresponds to best of 12 runs as the optimal lambda_1 value is chosen. However, despite the huge variability from run to run for some of the other algorithms (e.g., two runs of LS: 28%, 50%; two runs of UDFS: 29%, 70%), it seems they did not benefit from multiple runs. Do I understand correctly?

---

> > > > > ### Author Response · Authors · 2021-08-25
> > > > > **Response to Reviewer 4y4c. We tuned $\lambda_1$ only on validation data, and then reported our results on test data.**
> > > > >
> > > > > We sincerely appreciate you, reviewers, and AC for your valuable time in reviewing our responses to help improve the quality of our paper.
> > > > >
> > > > > For your question, we have responsed as follows.
> > > > >
> > > > >
> > > > > ---
> > > > > ---
> > > > >
> > > > >
> > > > > ## Question 1:
> > > > >
> > > > > > + *... However, I still have doubts regarding the experimental setup: the reported results for the proposed algorithm corresponds to best of 12 runs as the optimal $\lambda_1$ value is chosen. However, despite the huge variability from run to run for some of the other algorithms (e.g., two runs of LS: 28%, 50%; two runs of UDFS: 29%, 70%), it seems they did not benefit from multiple runs. Do I understand correctly?.*
> > > > >
> > > > >
> > > > > ## Response 1:
> > > > >
> > > > > + We are afraid that this could be a misunderstanding. We tuned $\lambda_1$ (out of 12 possible values listed in Footnote 6, Page 7) only on the validation set; after tuning $\lambda_1$, we reported the results of our algorithm on test data. As we responded to your **Question 2**, dated August 20 (the third paragraph): Once the tuning of $\lambda_1$ has been done on the validation data, it will not be changed on the test data. For easy or more general applicability of our algorithm, we tuned $\lambda_1$ only on the validation data of MNIST-Fashion (please see **Response 4** to **Question 4** of Reviewer 2), and then we used the tuned $\lambda_1$ on the test data of this dataset and 9 other datasets; see Line 241, Page 7.
> > > > >
> > > > > +  For other algorithms in comparison, we mainly followed the way in the literature, for example, the implementation of AEFS in Ref. [1], or the tuned parameters in the original papers, or the default values in the Python library `scikit-feature`, e.g., for LS (Ref. [24]).
> > > > >
> > > > > + Regarding the way of reporting results on test data, for a fair comparison, we also followed Ref. [1]. The results in Table 3 are the classification results on 10 datasets.
> > > > >
> > > > > + The two classification results you mentioned for LS and UDFS correspond respectively to Datasets 3 and 7. We believe it should be usual for a classifier to have performance variations on different datasets as pointed out in **Response 1** to your **Question 1** in the initial review.

---

> > > > > > ### Comment · Reviewer_4y4c · 2021-08-26
> > > > > > **tuning lambda_1**
> > > > > >
> > > > > > It seems any constant lambda_1 is enough for uniform stability according to Thm. 1. This argues for a small lambda_1 to optimize the feature selector cost. Indeed, hyperparameter tuning for lambda_1 produced a very small lambda_1 (1/2^7).
> > > > > > - I wonder why this optimization doesn’t produce even smaller lambda_1 values. Is it just numerical issues?
> > > > > > - If lambda_1=0, the proposed method would be identical to AEFS. (Am I correct?) Given that lambda_1 is very small, I’m not sure I understand the big performance difference between these two methods. Could you please comment on attuning ?
> > > > > >
> > > > > > Thanks.

---

> > > > > > > ### Comment · Area_Chair_7p8k · 2021-08-26
> > > > > > > **Lambda cannot be any constant**
> > > > > > >
> > > > > > > The bound in Eq. 4 is $O(\frac{1}{n} + \frac{1}{\lambda_1 n})$. If $\lambda_1 \to \infty$, the second term, $O(\frac{1}{\lambda_1 n})$, blows up. So the bound does _not_ say that any $\lambda_1$ is sufficient for stability. What is says is that some amount of regularization is sufficient for stability, but making $\lambda_1$ very large cannot improve the stability beyond $O(\frac{1}{n})$ (which is still sufficient for generalization).

---

> > > > > > > > ### Author Response · Authors · 2021-08-27
> > > > > > > > **Response to Reviewer 4y4c. Our proposed algorithm is different from AEFS.**
> > > > > > > >
> > > > > > > > We sincerely appreciate you, Reviewers, and AC for your valuable time in reviewing our responses and providing comments.
> > > > > > > >
> > > > > > > > For your questions, we have responded as follows.
> > > > > > > >
> > > > > > > >
> > > > > > > > ---
> > > > > > > > ---
> > > > > > > >
> > > > > > > >
> > > > > > > > ## Question 1:
> > > > > > > >
> > > > > > > > > *It seems any constant lambda_1 is enough for uniform stability according to Thm. 1. This argues for a small lambda_1 to optimize the feature selector cost. Indeed, hyperparameter tuning for lambda_1 produced a very small lambda_1 (1/2^7).*
> > > > > > > >
> > > > > > > > > + *I wonder why this optimization doesn’t produce even smaller lambda_1 values. Is it just numerical issues?.*
> > > > > > > >
> > > > > > > > ## Response 1:
> > > > > > > >
> > > > > > > > + We have read the comments of AC regarding the values of $\lambda_1$ and we agree with those insightful comments.
> > > > > > > >
> > > > > > > > + Regarding the question, the chosen value from tuning $\lambda_1$ on MNIST-Fashion was not a numerical issue. In our paper, $\lambda_1$ was determined from a set of 12 possible values on the validation data, as pointed out in our previous response to your **Question 1** dated August 24. If we search from a set of more values or tune $\lambda_1$ for each dataset individually (please see the first paragraph in **Response 1** to your **Question 1** dated August 24), we may have different $\lambda_1$ values and potentially better performance. But the chosen value already showed reasonable performance. Besides, for a regularization factor, we think a value of 1/2^7 = 0.0078 is in its normal range.
> > > > > > > >
> > > > > > > > $\quad$
> > > > > > > >
> > > > > > > > ---
> > > > > > > >
> > > > > > > >
> > > > > > > > ## Question 2:
> > > > > > > >
> > > > > > > > > + *If lambda_1=0, the proposed method would be identical to AEFS. (Am I correct?) Given that lambda_1 is very small, I’m not sure I understand the big performance difference between these two methods. Could you please comment on attuning ?*
> > > > > > > >
> > > > > > > > ## Response 2:
> > > > > > > > + AEFS is an interesting model; however, it is quite different from our algorithm:
> > > > > > > >
> > > > > > > >    + In our algorithm, the first term used a constraint that requires only the weights of top-$k$ features to be nonzero, which can be regarded as directly imposing an $\ell_0$ constraint for feature selection, but AEFS mainly depended on $\ell_{2,1}$ and Frobenius norms to get a sparse matrix to perform feature selection.
> > > > > > > >
> > > > > > > >   + Without the second term in our algorithm, we did not know of any stability guarantee; with the second term, our algorithm has the stability certificate.
> > > > > > > >
> > > > > > > >
> > > > > > > > + In our algorithm, we do not allow $\lambda_1=0$ (please see **Response 1** to **Question 1** of Reviewer 3).

---

> > > > > > > > > ### Comment · Reviewer_4y4c · 2021-08-31
> > > > > > > > > **thanks**
> > > > > > > > >
> > > > > > > > > No, a small **constant** lambda_1 should indeed be enough for asymptotic stability. In any case, asymptotic stability is not a concern for the experiments reported in the recon. error table.
> > > > > > > > >
> > > > > > > > > Thanks for patiently explaining — sorry, I confused the different autoencoder-based methods.

---

> > > > > > > > > > ### Author Response · Authors · 2021-09-01
> > > > > > > > > > **Response**
> > > > > > > > > >
> > > > > > > > > > Our pleasure to clarify these questions.

---

### Comment · Area_Chair_7p8k · 2021-08-27
**Question about baselines**

Hello authors,

Apologies for the late question. I'm hoping you can clarify some confusion happening in the reviewer discussion.

If I've understood the situation correctly, both the proposed method and certain baselines (e.g., AEFS) use an autoencoder (AE) architecture, and are therefore flexible in how they are instantiated; the encoder/decoder parts can either be linear models or deep NNs. In the reviews, there was some confusion as to whether the results of the proposed method were comparable to those of the baselines (which were taken from a reference [1]), since it was not clear whether all AE-based algorithms used same AE architecture -- specifically, whether the decoder was linear or a deep NN. This detail is important because the evaluation (test) metrics used a linear model for reconstruction error, and our concern is that an encoder trained for a nonlinear decoder may not work well later if paired with a linear decoder. (I hope I've captured the situation correctly.)

In your response, you say:

> We believe that the base architectures and metrics are reasonable or standard, since the architectures of algorithms in comparison, for instance, AEFS, used the same standard shallow autoencoder with a single hidden layer as Ref. [1].

Does "base architecture" refer to the encoder/decoder used in training, or the models used for evaluation? Does this mean that the baselines and proposed method use the exact same architecture?

Thank you for your time and patience.

---

> ### Author Response · Authors · 2021-08-28
> **Response to AC**
>
>
> We sincerely appreciate you for your valuable time in reviewing our responses, discussing,  and providing comments. We are pleased to clarify the uncertainty, and our responses are as follows.
>
>
> ---
> ---
>
>
> ## Question 1:
>
> > + *If I've understood the situation correctly, both the proposed method and certain baselines (e.g., AEFS) use an autoencoder (AE) architecture, and are therefore flexible in how they are instantiated; the encoder/decoder parts can either be linear models or deep NNs...since it was not clear whether all AE-based algorithms used same AE architecture -- specifically, whether the decoder was linear or a deep NN....*
>
> ## Response 1:
>
> + Yes.
> + Our decoder is linear and shallow, and it is the same as the baseline methods, such as CAE and AEFS, for which we used the original implementation in Ref. [1] for a fair comparison, as mentioned in our previous response (see **Response 1** to **Question 1** of Reviewer 1 dated August 20). For AgnoS-S, we used the codes from the original paper (Ref. [10]), whose implementation has also one hidden layer with the same number of parameters.
>
>
> $\quad$
>
> ---
> ---
>
> ## Question 2:
>
> >*Does "base architecture" refer to the encoder/decoder used in training, or the models used for evaluation? Does this mean that the baselines and proposed method use the exact same architecture?*
>
> ## Response 2:
>
> + "Base architecture" refers to the encoder/decoder used in training.
> + We used the same linear decoder as that in Ref. [1]. For the encoder, for a fair comparison, our algorithm used an AE having one hidden layer with the same number of neurons as other AE-based baselines in comparison. And we used the linear activation function for the encoder in the same way as CAE and AEFS in Ref. [1]. Besides, for feature selection, the first layer of the encoder in our global-NN used the slack variables in the same way as AgnoS-S in Ref. [10].

---

> > ### Comment · Area_Chair_7p8k · 2021-09-01
> > **Thanks**
> >
> > Thanks for the explanation!

---

> > > ### Author Response · Authors · 2021-09-02
> > > **Our pleasure. Thanks.**
> > >
> > > It was our pleasure to explain them. We sincerely appreciate your valuable time and effort in reviewing our responses, discussing, clarifying the uncertainties, and giving insightful comments. Thanks!

---

### Decision · Program_Chairs · 2021-09-27

**Decision:**

Accept (Poster)

**Comment:**

This paper proposes an unsupervised feature selection algorithm motivated by _algorithmic stability_, in the sense that the algorithm should not be overly sensitive to changes in the dataset. The algorithm consists of a feature _scorer_ and a feature _selector_, which uses the scorer to select features. These components can be instantiated as neural nets and trained jointly. The paper analyzes the algorithm's _uniform stability_ (the strongest notion of stability), proving that it is $O(1/n)$, which is sufficient for generalization. Experiments show that the algorithm performs well on multiple datasets, and confirms the theoretical claims about stability.

The reviews were mixed, but after much discussion, and back-and-forth with the authors, **I believe the correct decision is to accept.**

Let me first address some concerns raised by Reviewer 4y4c (paraphrased for brevity).

1. _The stability bound is uninformative because it is not fully controlled by the regularization parameter, $\lambda_1$; in particular, if over-regularized ($\lambda_1 \to \infty$), the bound does not go to zero._

Since the stability bound is $O(\frac{1}{n} + \frac{1}{\lambda_1 n})$, over-regularizing does not make it vanish. However, $O(1/n)$-uniform stability is sufficient for generalization, so I don't see this as an issue. Another reviewer also made a good point about this: "I have to remind myself that the role of the regularizer is not to control complexity. In particular, if the regularization term penalized the complexity of the model, then taking $\lambda_1 \to \infty$ should have vanishing stability, independent of $n$. But in their formulation, taking $\lambda_1$ to infinity would mean that only the feature score is effectively minimized and it is not surprising that $O(1/n)$ is what you might expect in this setting."

2. _According to the stability bound, $\lambda_1$ can be any constant (w.r.t. $n$) and guarantee stability. So why not just take $\lambda_1$ as small as possible? How then should one choose $\lambda_1$?_

While it is true that any constant $\lambda_1$ ensures $O(1/n)$ stability, and that this bound vanishes as $n \to \infty$, it is important to remember that what we ultimately care about is generalization error, and that we always have a finite amount of data. Given a certain data size, $n$, we want a value of $\lambda_1$ that balances stability (hence, generalization) with our ability to fit the data (i.e., reduce empirical risk). Determining this optimal value analytically is likely infeasible, so most people do it empirically via cross-validation (like you would any other hyper-parameter). Indeed, this is precisely what the authors do.

3. _The experiments borrow some results from prior work, and in doing so give the proposed method an advantage over some baselines. In particular, the baselines were optimized for a nonlinear decoder, while the proposed method uses a linear decoder, and the evaluation metrics involve linear decoders._

I believe this is simply a misunderstanding. The authors clarified in the discussion period that, "our decoder is linear and shallow, and it is the same as the baseline methods, such as CAE and AEFS, for which we used the original implementation in Ref. [1]." Thus, I think the comparison is fair.

Having settled these concerns, I don't see anything particularly wrong with the paper. The theory is sound; the experiments are sound; and the results are positive.

That said, why am I not making a stronger Accept recommendation?
1. For starters, the paper could use some polishing. The fact that there were so many misunderstandings indicates that the presentation could be improved. I myself find the notation used in the stability analysis to be a little overwhelming at times. But this can be fixed.
2. I don't find it all that surprising that the algorithm is uniformly stable, given that its objective function is strongly convex (under certain conditions on the hypothesis class). This is not to say that every paper must come to an unexpected conclusion, but that it seems straightforward to prove uniform stability in this case; you just need to make the right assumptions and then follow the proof techniques laid out in prior work. That being said, the main insight of the theory is that just optimizing a selector alone is insufficient to guarantee stability (and generalization); one must also optimize the scoring function for stability. That takeaway is valuable, even if the proof follows from prior work.
3. None of the reviews seemed all that excited about the paper. Not every paper has to be revolutionary, but at a competitive conference, such as NeurIPS, it helps to have at least one high-scoring review.

So, consider my recommendation "accept if possible."

Note for the authors: In Assumptions 1 & 2, the order of quantifiers seems off. They go, roughly, "$\forall x$, $\exists$ ..." This means that for any given $x$, there exists something (either a subset of the data or a constant) such that some property holds. But I think for the bound to hold uniformly what they really want is, "$\exists$ ... such that $\forall x$ ..."